# Transmission and dynamics of mother-infant gut viruses during pregnancy and early life

Sanzhima Garmaeva [1,14], Trishla Sinha [1,14], Anastasia Gulyaeva [1], Nataliia Kuzub[1], Johanne E. Spreckels [1], Sergio Andreu-Sánchez [1,2], Ranko Gacesa [1,3], Arnau Vich Vila [1,3], Siobhan Brushett [1,4], Marloes Kruk[1], Lifelines NEXT cohort study*, Jackie Dekens[1,5], Jan Sikkema[5], Folkert Kuipers [2,6], Andrey N. Shkoporov [7,8], Colin Hill [7,8], Sicco Scherjon[9], Cisca Wijmenga [1], Jingyuan Fu [1,2], Alexander Kurilshikov [1,15] & Alexandra Zhernakova [1,15] ✉

Early development of the gut ecosystem is crucial for lifelong health. While infant gut bacterial communities have been studied extensively, the infant gut virome remains under-explored. To study the development of the infant gut virome over time and the factors that shape it, we longitudinally assess the composition of gut viruses and their bacterial hosts in 30 women during and after pregnancy and in their 32 infants during their first year of life. Using shotgun metagenomic sequencing applied to dsDNA extracted from Virus-Like Particles (VLPs) and bacteria, we generate 205 VLP metaviromes and 322 total metagenomes. With this data, we show that while the maternal gut virome composition remains stable during late pregnancy and after birth, the infant gut virome is dynamic in the first year of life. Notably, infant gut viromes contain a higher abundance of active temperate phages compared to maternal gut viromes, which decreases over the first year of life. Moreover, we show that the feeding mode and place of delivery influence the gut virome composition of infants. Lastly, we provide evidence of co-transmission of viral and bacterial strains from mothers to infants, demonstrating that infants acquire some of their virome from their mother's gut.

The human early-life gut ecosystem has garnered much interest in recent years because of its links to health and disease later in life, but core aspects of its origin and development remain poorly understood[1]. Previous studies have characterised the development of the infant gut microbiome through the first 2–3 years of life, after which the gut microbiome reaches a state of high microbial richness and diversity that is similar to that of an adult[2–6]. While the focus of research thus far has been the developing gut bacteriome, a community of all bacteria inhabiting an ecosystem, the gut ecosystem also comprises viruses, archaea and eukaryotes. The role of these understudied microbiome

[1]Department of Genetics, University of Groningen, University Medical Center Groningen, Groningen, the Netherlands. [2]Department of Pediatrics, University of Groningen, University Medical Center Groningen, Groningen, the Netherlands. [3]Department of Gastroenterology and Hepatology, University of Groningen, University Medical Center Groningen, Groningen, the Netherlands. [4]Department of Health Sciences, University of Groningen, University Medical Center Groningen, Groningen, the Netherlands. [5]University Medical Center Groningen, Center for Development and Innovation, Groningen, Netherlands. [6]European Research Institute for the Biology of Ageing (ERIBA), University of Groningen, University Medical Center Groningen, Groningen, the Netherlands. [7]APC Microbiome Ireland, University College Cork, Cork, Ireland. [8]School of Microbiology, University College Cork, Cork, Ireland. [9]Department of Obstetrics and Gynecology, University of Groningen, University Medical Center Groningen, Groningen, the Netherlands. [14]These authors contributed equally: Sanzhima Garmaeva, Trishla Sinha. [15]These authors jointly supervised this work: Alexander Kurilshikov, Alexandra Zhernakova.*A list of authors and their affiliations appears at the end of the paper. ✉e-mail: a.zhernakova@umcg.nl

members in the early gut ecosystem is perceived to be very important, but their composition and development over time have received little attention.

Microbes from the maternal gut, skin and vaginal tract have been described as sources of the infant gut microbiota[7,8], and recent studies provide increasing support for the maternal gut bacterial reservoir as a key source of microbes transmitted from mothers to infants[9–15]. While mother-to-infant transmission of human viruses such as human immunodeficiency virus, cytomegalovirus and herpes simplex virus has been established in the context of maternal and infant morbidity[16], little is known about the transmission of bacteriophages (bacteria-infecting viruses) from the maternal to the infant gut. Studies of viral transmission and more generally, virome, a community of all viruses inhabiting an ecosystem, have been hindered by difficulties in isolating and annotating viruses[17]. As environmental studies have demonstrated that bacteriophages are key players in the modulation of bacterial communities[18,19], it is crucial to study them in the context of the developing human gut ecosystem as the bacterial community is established in the months following birth. To this end, a crucial Liang et al. study examining virus-like particle (VLP) data in 20 healthy infants provided evidence that the bacteriophages colonising the infant gut arise from excisions from pioneering infant gut bacteria[20]. However, the source of these pioneering bacteriophages, as well as their hosts and possible roots in the maternal gut ecosystem, have remained elusive. In the limited number of studies to examine both maternal and neonatal samples, infant faecal samples collected within 4 days post-partum shared 15% of their viruses with the respective maternal samples[21]. Another study, which focused on bifidobacterial phages, showed that infants acquire *Bifidobacterium* phages from their mother[7]. Contrastingly, a recent study concluded that maternal exposure does not directly impact the development of the infant gut virome[22].

In this study, we investigate the composition of the maternal and infant gut virome during pregnancy and the first year after birth, associate it to host and environmental factors and investigate whether infants and their mothers share some viruses. To do so, in 30 mothers and their 32 infants, including two twin pairs, we sequenced 322 total metagenomes, representing the total sum of all genomes obtained by isolating total microbial DNA from stool and 205 VLP metaviromes, representing the total sum of all viral genomes, obtained using a VLP-enrichment isolation protocol. We found that the composition of the infant virome is highly dynamic during the first year of life and remains different from the maternal virome at 1 year of age. In contrast, the maternal virome is relatively stable during late pregnancy and after birth. At early timepoints, the infant viromes are dominated by active temperate bacteriophages, the abundance of which decreases over time. Lastly, we show evidence for the transmission of viruses from mothers to infants in related mother-infant pairs, indicating that infants derive some of their virome from their mothers.

## Results

### Study population
We profiled the gut microbiome (primarily referred to as the bacteriome) in 322 total metagenome samples and the double-strand DNA (dsDNA) gut virome in 205 VLP metavirome samples from 30 mothers and their 32 term-born infants (including 2 twin pairs) collected longitudinally from pregnancy to 12 months after birth (Fig. 1a; Supplementary Fig. 1a, b). The infants had a median birth weight of 3700 g (range: 2462–5055 g). A significant majority of the children, 87.5% (28 infants), were delivered vaginally, and 28.1% (9 infants) were born at home (Supplementary Fig. 1c). Median maternal age at childbirth was 32 years (range: 24–40 years). Breastfeeding behaviour showed a gradual decline over the initial 3 months of life (Supplementary Fig. 1d, e). In the first month, half of the infants were exclusively breastfed, falling slightly to 43.8% in the second month

and to 33.3% by the third month. The summary statistics of the study population is presented in Supplementary Data 1. After processing all samples through our bacteriome and virome annotation pipelines (Methods), we characterised the maternal gut virome during pregnancy, at birth and in the first 3 months after birth and the infant gut virome over the first year of life, along with the predicted bacterial hosts. We then related the infant virome composition with feeding mode and birth-related factors and described the infant virome acquisition by means of transmission from the mother.

### The infant gut virome is dynamic during the first year of life
To characterise the infant virome directly after birth, we first attempted to sequence the VLP metavirome and total metagenome from meconium. However, none of the 17 virome DNA isolations from meconium could be sequenced, and only 7% ($n = 2$) of the total metagenomic samples yielded viable sequencing data (microbial read depth > 5 million reads). These outcomes highlight the remarkably low microbial biomass present in meconium samples, further reinforcing the prevailing consensus that the gut remains sterile before birth[23,24].

To investigate temporal changes in the infant virome after birth across the first year of life, we sequenced VLP metaviromes from five timepoints (months 1, 2, 3, 6 and 12) and total metagenomes (mainly comprising bacteria) from six timepoints (months 1, 2, 3, 6, 9 and 12) from faecal samples from 32 infants. Using all 205 VLP metaviromes from mothers and infants, we reconstructed 102,270 virus operational taxonomic units (vOTUs, i.e. species-rank virus groups defined using standard thresholds of 95% average nucleotide identity (ANI) over 85% alignment fraction (relative to the shorter sequence)[25]). Of these, 10.7% of vOTUs were infant-specific (detected only in infants), and 76.1% were mother-specific. The average number of vOTUs in infants (889, 95% confidence interval (CI, Methods): [719, 1060]) was lower than in mothers (5182, 95%CI: [4294, 6096]; $p$ value = 1.6e-07, beta = −4304; Supplementary Fig. 1f; Supplementary Data 2). Over the course of infancy, the average number of vOTUs in infants increased from an average of 675 (95% CI: [299.0, 1234.9]) at the age of 1 month to an average of 1352.9 (95% CI: [977.0, 1732.2]; Supplementary Fig. 1f) at 12 months. The overall composition of both viruses and bacteria in the infants underwent significant changes over time (Bray-Curtis dissimilarity, NMDS with one dimension, $p$ value = 1.1e-03, beta = −5.3e-04, for virome, and $p$ value = 2.0e-07, beta = −1.8e-03, for bacteriome; Supplementary Data 3, 4), moving towards a mother-like state (Fig. 1b, c). Both the overall virome and bacteriome composition were significantly different between mothers and infants ($p$ value = 1.5e-10, beta = −0.2, for virome, and $p$ value = 3.6e-36, beta = −1.3, for bacteriome; Supplementary Data 5, 6).

We investigated the temporal changes in the maternal virome at five different timepoints (month 7 of pregnancy, birth and months 1, 2 and 3 after birth) and observed that the maternal virome did not change significantly with time ($p = 0.6$, beta = −0.005; Fig. 1d; Supplementary Data 7). The maternal microbiome also did not change significantly during late pregnancy and the examined postpartum period, but we did observe a significant change in the bacterial composition from month 3 to month 7 of pregnancy ($p$ value = 2.3e-4, beta=0.02, Fig. 1e; Supplementary Data 8), which agrees with previous findings[26,27].

Gut viral and bacterial alpha diversity were higher in mothers than in infants in the first year of life ($p$ value = 1.1e-09, beta=2.5, for virome; $p$ value = 5.9e-29, beta = 1.8, for bacteriome; Supplementary Data 9, 10; Fig. 2a, b). The alpha diversity of both the infant gut virome and bacteriome increased with age, approximating but not reaching the alpha diversity of mothers by 1 year of life ($p$ value = 4.1e-4, beta=4.7e-03, for virome; $p$ value = 1.6e-12, beta=2.7e-03, for bacteriome; Fig. 2a, b; Supplementary Data 11, 12). Even at the age of 1 year, there was still a significant difference between the viromes diversity of infants and their mothers (two-sided Wilcoxon signed-rank test, $p$ value = 1.7e-3). The alpha diversity of the maternal gut virome and bacteriome showed

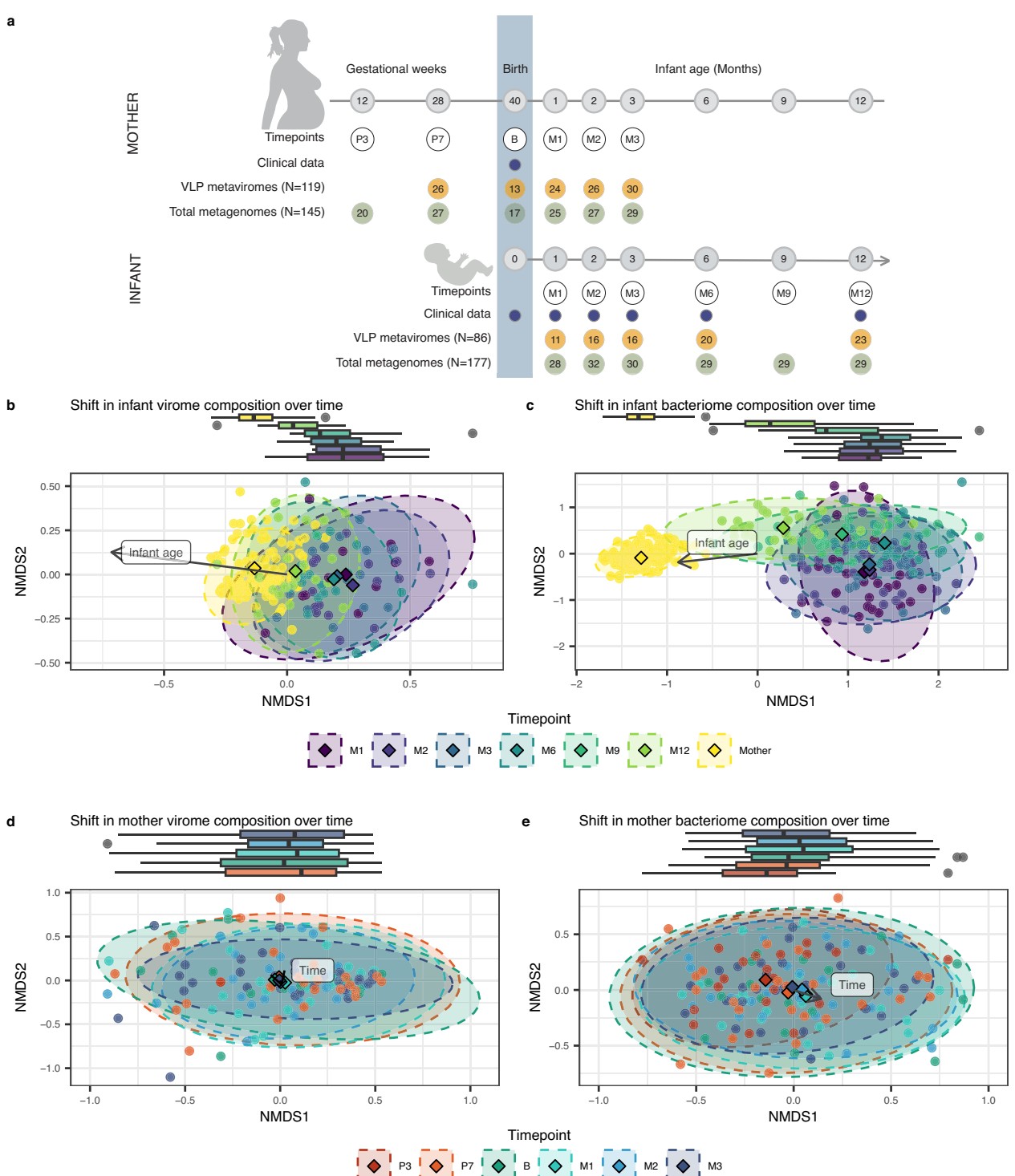

**b** Shift in infant virome composition over time

**c** Shift in infant bacteriome composition over time

**d** Shift in mother virome composition over time

**e** Shift in mother bacteriome composition over time

no significant changes, remaining stable throughout pregnancy and after delivery (*p* value = 1, beta=4e-04, for virome; *p* value = 0.3, beta=1.1e-02, for bacteriome; Fig. 2a, b; Supplementary Data 13, 14).

We then sought to determine whether the increase in viral alpha diversity in infants over time was a result of retention of the initial coloniser vOTUs paired with the introduction of new species over time. To address this, we investigated the persistence of the viruses initially detected in the infant gut at month 1. By month 2, an average of 37.8% (95% CI: [19.1, 58.8]) of the vOTUs present at month 1 were retained, and these retained vOTUs on average accounted for 42.2% of vOTUs detected at month 2 (Fig. 2c). Both the proportion of retained month 1

vOTUs and the percentage of richness these vOTUs occupied per sample decreased significantly over time, along with their relative abundance (*p* value = 1.3e-02, beta = −2.8e-02; *p* value = 9.1e-04, beta=3.4e-02; *p* value = 7.0e-3, beta = −3.6, Fig. 2c, d; Supplementary Data 15). In contrast, the maternal virome remained relatively stable compared to that of infants, as birth and postpartum samples, on average, retained 42.9% of the vOTUs detected at month 7 of pregnancy, with the relative abundance of these retained vOTUs accounting for 74.7% on average (Fig. 2e, f). We observed similar trends for the bacteriomes, i.e., a dynamic infant bacteriome over the first year and a stable maternal bacteriome (Supplementary Fig. 1g–j). In summary, the

**Fig. 1 | Overall composition of the infant and maternal gut virome and bacteriome over the first year of life and during pregnancy. a** Timeline and availability of infant and maternal total metagenomes and virus-like particle (VLP) metaviromes. Faecal samples were collected at gestational weeks 12 and 28, at birth, and months 1, 2, 3, 6, 9 and 12 after birth (numbers in grey circles). Timepoints are displayed in black circles. Blue circles denote the availability of clinical data per timepoint (Supplementary Data 1). Numbers in green and yellow circles indicate the number of samples per timepoint. Non-metric multidimensional scaling (NMDS) analysis based on Bray-Curtis dissimilarity calculated at the level of (**b**) vOTUs, (**c**) bacterial species, (**d**) mother vOTUs and (**e**) mother bacterial species. Linear mixed-effects models (LMM) showed the significant effect of timepoint on NMDS 1 in infants at the level of vOTUs and bacterial species (*p* value = 1.1e-03 & *p* value = 2.0e-07), as well as the significant effect of sample type (mother or infant) on NMDS 1 at the level of vOTUs and bacterial species (*p* value = 1.5e-10 & *p* value = 3.6e-36). LMM showed no significant effect of timepoint on NMDS 1 at the level of mother vOTUs (*p* = 0.6) and a significant effect of timepoint on NMDS1 at the level of mother bacterial species (*p* value = 2.3e-4). Sample sizes for mothers and infants at each timepoint are shown in (**a**). In all graphs, each point represents one sample, and different colours indicate samples from various timepoints, and in (**b**), (**c**), mother samples are shown in yellow. The distance between points reflects the level of dissimilarity. Centroids of sample clusters from the same timepoints are shown as diamonds in respective colours. Ellipses of respective colours represent 95% confidence regions for each timepoint, assuming a multivariate t-distribution of the data points. Boxed labels with arrows indicate significant vectors determined by fitting them onto ordinations. The boxplots on top of NMDS plots depict the distribution of NMDS1 per timepoint. Each boxplot visualises the median, hinges (25th and 75th percentiles), and whiskers extending up to 1.5 times the interquartile range from the hinges. Source data are provided as a Source Data file.

infant gut virome and bacteriome were highly dynamic during the first year of life, whereas the maternal gut virome and bacteriome were relatively stable.

Previous studies have demonstrated that adults possess a unique and stable collections of viruses, termed the personal persistent virome (PPV), that plays a crucial role in shaping the diversity and stability of the virome[28]. Considering the highly dynamic nature of the infant gut, we investigated whether the development of the PPV begins during infancy. For this analysis, we focused on the 14 infants for whom at least three stool sample VLP metaviromes were available and categorised the vOTUs present in these infants into PPV, i.e., vOTUs present in ≥75% of an individual's samples. The remaining, much-less-stable part of the virome can include phages of low abundance, those infecting transient microbiota members and plant viruses of dietary origin, which are together termed the transiently detected virome (TDV)[28]. We defined the TDV as those present in <75% of an individual's samples. In infants, PPVs were individual-specific, as 50.5% of all infant PPVs (*n* = 1054) were attributed only to a single infant. PPV in infants accounted for an average of 16.4% (95% CI: [11.7, 21.9]) of the vOTUs present per sample (Fig. 2g), whereas the TDV accounted for an average of 83.6% (95% CI: [78.5, 88.0], Fig. 2g). The average cumulative relative abundance of the PPV per sample was 35.1% (95% CI: [26.6; 44.6]) and lower than that of the TDV (*p* value = 8.8e06, beta = −29.8; Fig. 2h; Supplementary Data 16). However, it must be noted that infant viromes showed a large amount of inter-individual variation in the fractions occupied by PPVs and TDVs. In contrast to infants, the PPVs in mothers accounted for an average of 43.8% (95% CI: [39.2, 48.0]) of the vOTUs detected per sample, with the cumulative relative abundance accounting for an average of 72.3% per sample (Supplementary Fig. 2c, d).

Similar to the infant gut viromes, transiently detected bacteriomes (TDB) were the most prevalent in the infant gut bacteriomes (mean 67.6% per sample, *p* value = 2.1e-62, beta=35.2, Supplementary Fig. 2a; Supplementary Data 17). However, the cumulative relative abundance of personal persistent bacteriomes (PPBs) reaching on average 68.4% (95% CI: [63.5, 73.1]) outnumbered the relative abundance of TDB on average accounting for 31.8% (95% CI: [26.9, 36.8]; *p* value = 3.4e-23, beta=36.6; Supplementary Data 18). The maternal bacteriome, like the maternal virome, showed remarkable conservation and stability. It was dominated by PPB at both the richness (*p* value = 2.1e-131, beta = −53.4; Supplementary Fig. 2e; Supplementary Data 19) and abundance (*p* value = 7.5e-238, beta = −92.7; Supplementary Fig. 2f; Supplementary Data 20) levels. We thus concluded that the infant gut ecosystem contains persistent viruses, but they do not fully define the infant gut virome due to the large variability and high virus turnover with time.

### Virus−host interactions in the maternal and infant gut

We next predicted bacterial hosts for viruses based on vOTU representatives using the iPHoP framework[29]. We could assign 85,135 (83.3%)

vOTUs to their 4572 bacterial hosts at the species level and 68,299 (66.8%) vOTUs to 826 host genera (Fig. 2i, j; Methods). We then observed that the composition of the virome based on its predicted hosts closely resembles the bacterial composition and that the dynamics of the relative abundance of viruses closely mirror that of the host genera (Supplementary Fig. 3a-d). For example, over the first year of life in infants, we observe an increase in the relative abundance of bacteriophages predicted to infect the genus *Faecalibacterium* (FDR = 2.1e-12, beta=0.5; Supplementary Fig. 3c;) and a decrease in bacteriophages infecting bacteria from *Bifidobacterium* (FDR = 1.8e-4, beta = −0.3; Supplementary Fig. 3c) and *Klebsiella* (FDR=3e-04, beta = −0.2; Supplementary Fig. 3c; Supplementary Data 21) genera. We also observed similar trends for the abundance of corresponding bacteria (Supplementary Fig. 3b, d; Supplementary Data 22). We next checked if the predicted hosts of the viruses within PPVs were also, in fact, found in the PPBs, and we found that the overlap between predicted hosts of PPVs (210 species) and taxonomy of PPBs (69 species) was 35 species (Supplementary Fig. 3e). We noticed that despite the individual specificity of PPVs at the vOTU level, the majority of viruses within PPVs were predicted to infect bacterial species from the genus *Bacteroides* (56.3%) and *Phocaeicola* (10.1%).

To further establish the bacteria-virus dynamics, we focused on temperate bacteriophages because they have the ability to integrate their DNA into the host bacterium's genome, potentially influencing the bacterial phenotype and contributing to long-term interactions between bacteria and viruses. The relative abundance of active temperate bacteriophages (detected in VLP metaviromes) in infants was higher than that in mothers (*p* value = 1e-09, beta=22.2; Fig. 3a; Supplementary Data 23). The average relative abundance of temperate phages was high in the first 3 months (45.9%, 95% CI: [36.6, 55.4]) and decreased drastically by 6 months after birth (22.6%, 95% CI: [13.3, 31.8]; *p* value = 2.9e-04, beta = −8.7e-02; Fig. 3a; Supplementary Data 24). By 12 months, the relative abundance of active temperate bacteriophages was only slightly higher than that observed in maternal samples (on average 20.0%, 95% CI: [13.4, 27.4] in infants vs on average 11.0%, 95% CI: [9.8, 12.3] in mothers; *p* value = 0.045; two-sided Wilcoxon signed-rank test). Given that the majority of temperate bacteriophages in the adult gut are in the form of prophages[28], we compared the percentage of temperate bacteriophages in mothers and infants using the vOTUs detected in MGS metaviromes, representing the total sum of all viral genomes in total metagenomes. MGS metaviromes were created by aligning the reads from total metagenomes to vOTU database reconstructed from VLP metaviromes, thus representing the prophage-inclusive temperate phage content of the virome. In MGS metaviromes, we again observed that the prophage-accounting relative abundance of temperate bacteriophages was slightly higher in infants than in mothers (*p* value = 0.04, beta=2.1; Supplementary Fig. 4a; Supplementary Data 25). Overall, our findings demonstrate a higher relative abundance of temperate bacteriophages, both in their active form and as prophages, in infants compared to mothers,

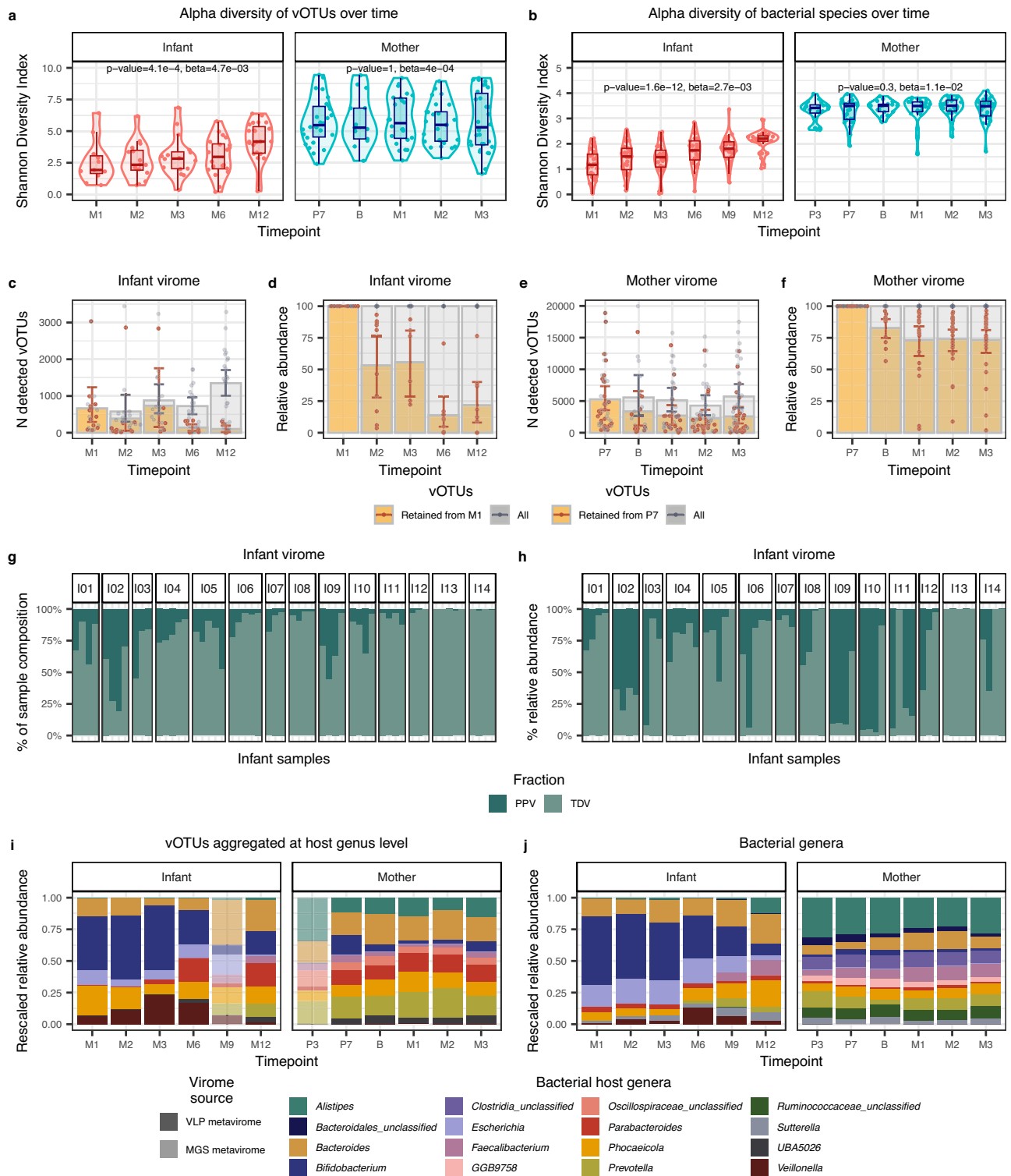

indicating their large potential impact on the dynamic interplay between bacteria and viruses in the infant gut.

## Infant gut virome composition is influenced by infant feeding mode and place of delivery

We next explored the effects of maternal age, infant sex, place of birth, birthweight, feeding mode and gestational age on infant gut viral and bacterial composition. The alpha diversity of vOTUs was associated with infant feeding mode throughout the first year of life, with exclusively formula-fed infants showing higher alpha diversity than

breastfed infants ($p$ value = 0.02, beta=0.9, Fig. 3b; Supplementary Data 26). This effect held true even after correcting for bacterial diversity ($p$ value = 0.01, beta=0.8; Supplementary Data 27). Feeding mode was also associated with the richness of active temperate bacteriophages, as exclusively formula-fed infants consistently showed a higher richness of temperate bacteriophages over time compared to breastfed infants ($p$ value = 3.0e-3, beta=45.4; Fig. 3c; Supplementary Data 28), also after correcting for vOTUs richness ($p$ value = 0.01, beta=15.0; Supplementary Data 29). The richness of temperate phages in MGS metaviromes also showed a significant association with

**Fig. 2 | Longitudinal dynamics of gut virome and bacteriome in infants and mothers. a, b** Alpha diversity, as measured by the Shannon Diversity Index, of (**a**), vOTUs and (**b**), bacterial species in mothers and infants over time. Boxplot colours represent infant (coral) or maternal (blue) samples. Boxplots visualise the median, hinges (25th and 75th percentiles), and whiskers extending up to 1.5 times the interquartile range from the hinges. Linear mixed-effects models showed a significant effect of timepoint on infant's viral ($p$ value = 4.1e-4) and bacterial diversities ($p$ value = 1.6e-12), while timepoint did not have a significant effect on mothers' viral ($p$ value = 1) and bacterial diversities ($p$ value = 0.3). Sample sizes are detailed in Fig. 1a. **c**, Number and **d**, relative abundance of vOTUs retained from month 1 in $n$ = 11 infants at months 2 ($n$ = 10), 3 ($n$ = 6), 6 ($n$ = 10), and 12 ($n$ = 9) after birth. **e**, Number and **f**, the relative abundance of vOTUs retained from the 7th month of pregnancy in $n$ = 26 mothers at birth ($n$ = 11) and months 1 ($n$ = 21), two ($n$ = 23) and three ($n$ = 26) after delivery. In (**c**)–(**f**), bar colours show retained (yellow) and all vOTUs (grey) per sample. The 95% confidence intervals (CI) for the

means are depicted in brown and grey whiskers for retained and all vOTUs, respectively. **g** Percentage of infant VLP metaviromes composition and (**h**) the cumulative relative abundance of vOTUs classified into biome fractions based on vOTUs persistence in 14 infants with at least three longitudinal samples. Personal persistent viromes (PPV, shown in deep teal) include vOTUs present in >= 75% of an individual's samples. Transiently detected viromes (TDV, shown in muted seafoam) include vOTUs present in less than 75% of the samples of an individual. **i, j** Stacked bar plots showing the top-10 most abundant vOTU aggregates by their bacterial hosts at the genus level and (**j**) top-10 abundant bacterial genera in infants and mothers. The relative abundance of vOTUs and bacterial genera was rescaled to fit to 1 after removing low abundant and unassigned groups. Translucent stacked bars indicate the MGS metaviromes for the timepoints that were unavailable for the VLP metavirome profiling. Opaque stacked bars indicate the VLP metaviromes. Source data are provided as a Source Data file.

formula feeding even after correction for bacterial richness ($p$ value = 0.02, beta=53.5; Supplementary Fig. 4b; Supplementary Data 30). We then looked at the effect of feeding mode on vOTUs aggregated on the basis of their host. Intriguingly, for some phages there are indications that the association can be attributed to the phage itself and not just its host. Compared to breastfed infants, exclusively formula-fed infants had more active temperate phages of bacteria from the genus *Bacteroides*, with the most significant association belonging to *Bacteroides fragilis* (FDR = 2.1e-3, beta=1.4; Fig. 3d), *Phocaeicola vulgatus* (FDR = 0.01, beta=1.5), and *Bacteroides caccae* (FDR = 0.02, beta=0.7), even after correction for the abundance of their bacterial hosts (Supplementary Data 31, 32). When corrected for both the host abundance and the estimated number of prophages from MGS metaviromes, the differential prevalence of active temperate phages of *B. fragilis* remained significantly higher in formula-fed infants (FDR = 0.02, beta=1.1; Supplementary Data 34). This suggests that formula feeding might be associated with the induction of bacteriophages in *B. fragilis*, independent of changes in bacterial abundances.

Since only two mothers gave birth to their infants by caesarean section, we did not have sufficient power to explore the effect of birth mode on virome composition (Supplementary Fig. 1c). However, as 28% of the infants were born by vaginal delivery at home, we investigated if home versus hospital delivery was associated with specific vOTUs aggregated by the microbial host or microbial taxa themselves. Here we observed that the bacterial species *Akkermansia muciniphila* was more abundant in infants born at home compared to those born at the hospital (FDR = 0.04, beta=2.5; Supplementary Fig. 4c; Supplementary Data 35). Concomitantly, the phages of *A. muciniphila* were also more abundant in infants delivered at home ($p$ = 0.02, beta=1.5; Supplementary Fig. 4d, e; Supplementary Data 36). This shows the importance of accounting for the birth environment when considering the early development of the infant virome and its interactions with its host(s).

### Infants can acquire gut viruses and bacterial hosts from the maternal gut

Despite the large difference in adult and infant gut microbes, related mother-infant pairs have previously been shown to share gut bacterial species[11,12], but limited information is available about sharing of viruses between maternal and infant guts. We therefore investigated if the gut ecosystem of mothers and their infants harbours the same viruses. We first compared the percentage of infant vOTUs that were shared with the pooled pre-birth (pregnancy month 7, Birth) and pooled post-birth (Month 1, 3) maternal samples. Infants, on average, shared a higher percentage of their vOTUs with post-birth maternal samples (32.3%

compared to the pre-birth maternal samples (26.6%) across all infant timepoints ($p$ = 0.04, beta=3.3e-02; Fig. 3e; Supplementary Data 37). As pioneer viruses in the infant gut are thought to be primarily temperate phages induced from the first gut bacterial colonisers[20], we next assessed the sharedness of maternal to infant vOTUs while accounting for prophages detected only in MGS metaviromes. Notably, sharedness increased significantly when considering prophages in both the maternal and infant gut ($p$ = 0.001; mean increase of 4.9%; Fig. 3f; Supplementary Fig. 4f; Supplementary Data 38). On average, the relative abundance of vOTUs shared with maternal gut virome in infants was 32.7% (95% CI: [28.1, 37.4]). We then sought to explore whether infant feeding mode, place of delivery and infant gestational age influenced the percentage of shared vOTUs between mother and infant, but this was not the case. These findings suggest that sharing of vOTUs between mothers and infants is more likely attributed to cohabitation rather than direct seeding of these viruses during birth. Additionally, the higher degree of sharedness observed when considering the MGS metavirome provides support for the notion that the presence of shared bacteria containing prophages contributes to the colonisation process in infants.

We next sought to see if there were cases of virus strain-sharing within mother-infant pairs. To do so, we selected vOTUs that were shared between mother and infant and passed strict cut-offs for completeness and coverage (Methods), resulting in 51 vOTUs for downstream analysis. For these 51 vOTUs, we reconstructed consensus sequences from quality-trimmed reads aligned to vOTUs from MGS and VLP metaviromes and calculated pairwise genetic distances (Kimura) between consensus sequences corresponding to the same vOTUs. We compared these genetic distances between the viruses shared across an infant and their own mother versus unrelated mothers. We found that, for 28 of the 51 vOTUs (55%), the genetic distance between related mother-infant sample pairs was significantly lower than that of unrelated mother-infant sample pairs (Fig. 4a). We then defined strain-sharing using distance cut-offs estimated assuming strains retention in longitudinal samples (Methods, Supplementary Fig. 4g). In 26 of these 28 viruses, we observed 841 strain transmission events between samples from related mothers and infants (Methods). Of the 26 transmitted viruses, 23 were shared with higher frequency within related mother-infant pairs compared to unrelated pairs (FDR < 0.05; Fig. 4b; Supplementary Data 39). Seven of these were found among PPVs in 50% of infants with more than 3 timepoints available. These persistent transmitted colonizers were predicted to infect bacteria from genera *Phocaeicola*, *Bacteroides* and *Parabacteroides*.

Next, we explored the transmission of the predicted bacterial hosts of the shared viruses. For the 26 transmitted viruses, we constructed 37 strains of their 29 bacterial hosts in both maternal and infant faecal samples (see Methods). Our findings indicate that, for 26 of the 30 (86.7%) reconstructed bacterial host strains present in

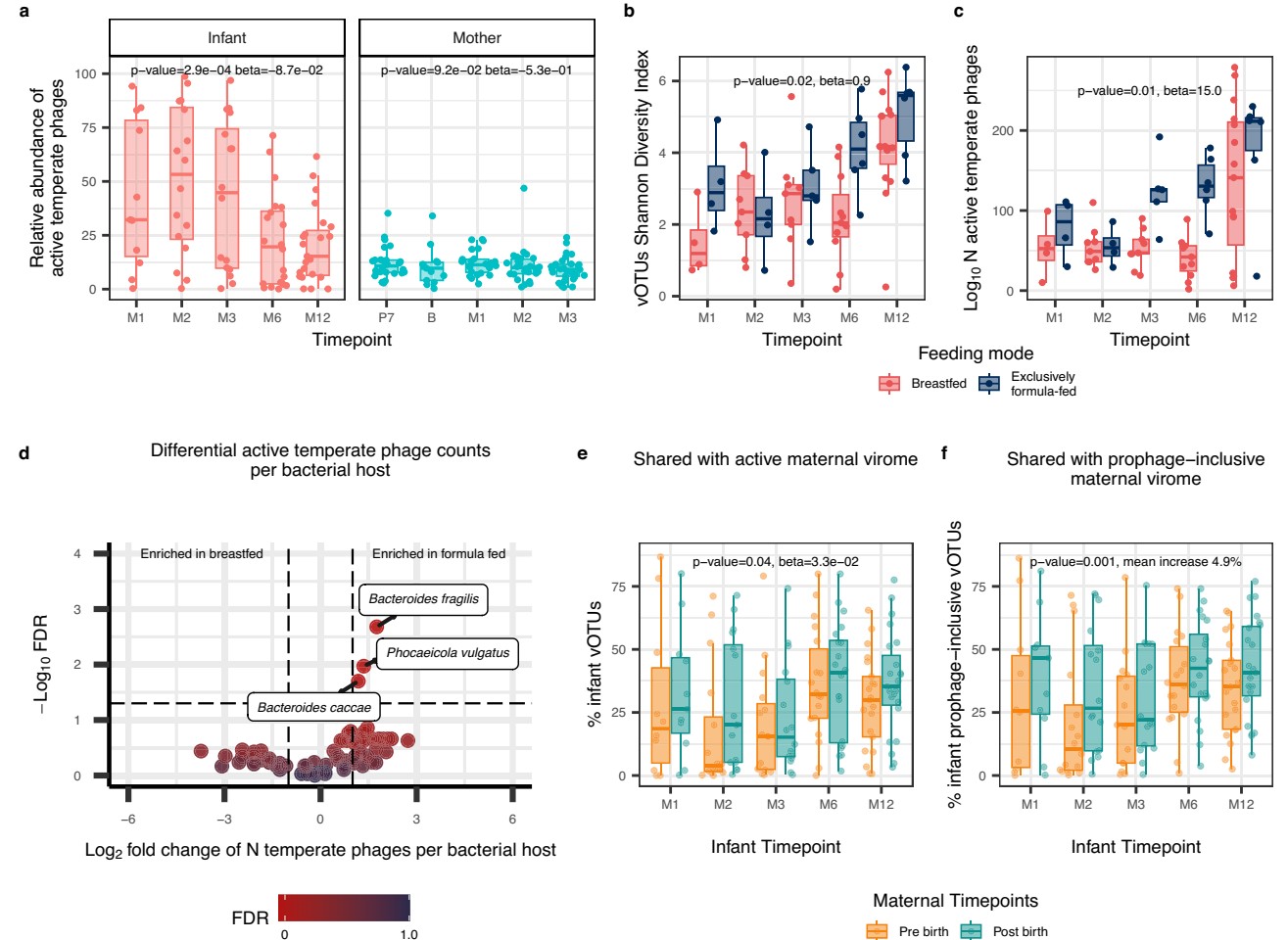

**Fig. 3 | Infant gut virome in relation to diet and maternal factors over time.**
**a** Relative abundance of active temperate phages in the infants and mothers over time in VLP metaviromes. Boxplot colours indicate infant (coral) or maternal (blue) samples. Linear mixed-effects model (LMM) showed a significant decrease in infant's active temperate phages abundance over the first year (p value = 2.9e-04), while that of mothers remained stable (p value = 9.2e-02). Sample sizes are detailed in Fig. 1a. **b** Alpha diversity, as measured by the Shannon Diversity Index, of vOTUs, and (**c**) Richness of active temperate phages in infants by feeding mode. Boxplot colours indicate the infant feeding type during sample collection: breastfed (red) and exclusively formula-fed (blue). LMM showed the significant effect of feeding mode on vOTUs diversity (p value = 0.02) and active temperate phage richness (p value = 0.01). **d** Volcano plot for the LMM associations between infant feeding mode and active temperate phage vOTU count per bacterial host, corrected for multiple testing (Benjamini-Hochberg). Dot colour represents the FDR significance. Boxed labels show the host species names for those vOTU aggregates whose count was significantly different between feeding modes (FDR < 0.05). The horizontal dashed line indicates an FDR cut-off of 0.05, and vertical dashed lines indicate a log-fold change cut-off of −1 or 1. For (**b**)–(**d**), data from n = 29 infants with available VLP metaviromes and feeding mode information at 5 timepoints is used. n(M1) = 8, n(M2) = 13, n(M3) = 14, n(M6) = 17, n(M12) = 21. **e** Percentage of infant vOTUs shared with pre-birth (orange) and post-birth (green) active maternal virome. **f** Percentage of sharedness of infant vOTUs with pre-birth (orange) and post-birth (green) maternal samples, taking into account the prophages (based on the presence of temperate phages in MGS metaviromes). In (**e**) and (**f**), sample sizes for infants at each timepoint are shown in Fig. 1a. Maternal samples were divided into two groups: pre-birth, n(P7) = 26, n(B) = 13, and post-birth n(M1) = 24, n(M3) = 30. LMM showed that infants shared more vOTUs with post-birth maternal active virome (p value = 0.04). Including prophages yielded a significant increase in the shared vOTUs (p value = 0.001). In (**a**)–(**c**), (**e**) and (**f**), boxplots visualise the median, hinges (25th and 75th percentiles), and whiskers extending up to 1.5 times the inter-quartile. Source data are provided as a Source Data file.

both mother and infant, the distances between related mother-infant pairs were lower than those observed between unrelated mother-infant pairs (FDR < 0.05; Fig. 4c). Of those 26 bacterial strains, 24 were shared with higher frequency within related mother-infant pairs compared to unrelated pairs (FDR < 0.05; Fig. 4d; Supplementary Fig. 5; Supplementary Data 39). These bacterial strains mostly belong to the genera *Alistipes, Bacteroides, Bifidobacterium, Faecalibacterium, Parabacteroides, Phocaeicola* and *Sutterella*.

To establish whether viral transmission occurred during or after birth, we investigated if there was a difference in viral strain-sharing between infant samples (at all timepoints) and maternal pre-birth (pregnancy month 7, birth) versus post-birth (month 1, month 3) samples. Here, we found one significant difference in viral strain-sharing between pre-birth and post-birth samples (FDR < 0.05;

Supplementary Data 39). In concordance with this, the host of this virus, *Parabacteroides distasonis*, was also shared with a higher frequency between infant samples and post-birth maternal samples as compared to pre-birth maternal samples (p value < 0.05; Supplementary Data 39), suggesting a higher probability that this bacterium and its phage were transmitted after birth.

Subsequently, we tested whether bacteriophages were preferably co-transmitted alongside their bacterial hosts, as opposed to other bacteria, by correlating their strain-sharing events in concurrent samples (Methods). Our observations revealed that bacteriophages were predominantly co-transmitted in conjunction with their bacterial hosts (p value = 0.01; Fig. 5a, b; Supplementary Data 40). An example of one such bacteriophage is L85266_LS0, whose host *Bacteroides uniformis* shows a very similar topological pattern in its phylogenetic

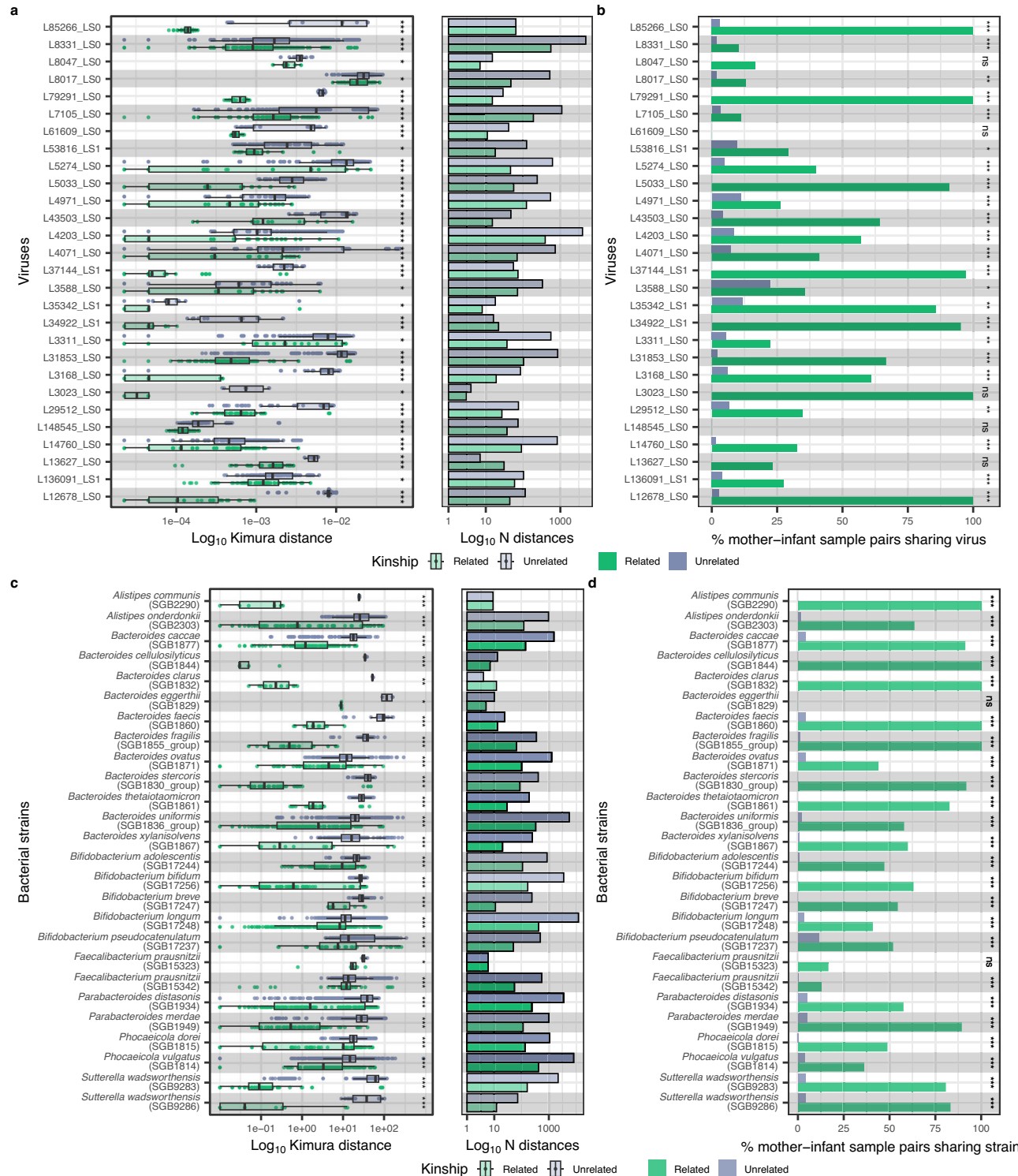

**Fig. 4 | Sharing of gut bacterial and viral strains between mothers and infants.**
Log scaled Kimura 2-parameter distances of (**a**) viruses and (**c**) bacterial strains with significantly lower genetic distances in related mother-infant pairs (shown in the colour green) compared to unrelated mother-infant pairs (shown in the colour periwinkle). Significance of the distance comparison was determined using one-sided Wilcoxon rank sum test, conducted through a permutation approach with $n = 1000$ iterations. The panels on the right of (**a**) and (**c**) indicate the total number of pairwise distances for the related and unrelated mother-infant pairs for the virus/ bacterium. Percentage of mother-infant sample pairs sharing the same strain of (**b**) viruses and (**d**) bacterial species as defined by the within-individual strain variation cut-off. Significance of the strain-sharing enrichment in related mother-infant pairs was determined using one-sided Fisher's exact test. Y-axis of (**a**) and (**b**) contain the labels of viruses composed of scaffold length (L) and predicted lifestyle (LS, followed by 0 for virulent and 1 for temperate phages). All boxplots visualise the median, hinges (25th and 75th percentiles), and whiskers extending up to 1.5 times the interquartile range from the hinges. All $p$ values were corrected for multiple testing using the Benjamini-Hochberg method. Asterisks denote FDR significance, *FDR < 0.050; **FDR < 0.010; ***FDR < 0.001; ns=not significant. Exact $p$ values, FDR-values and sample size per virus or bacterium can be found in Supplementary Data 39. Source data are provided as a Source Data file.

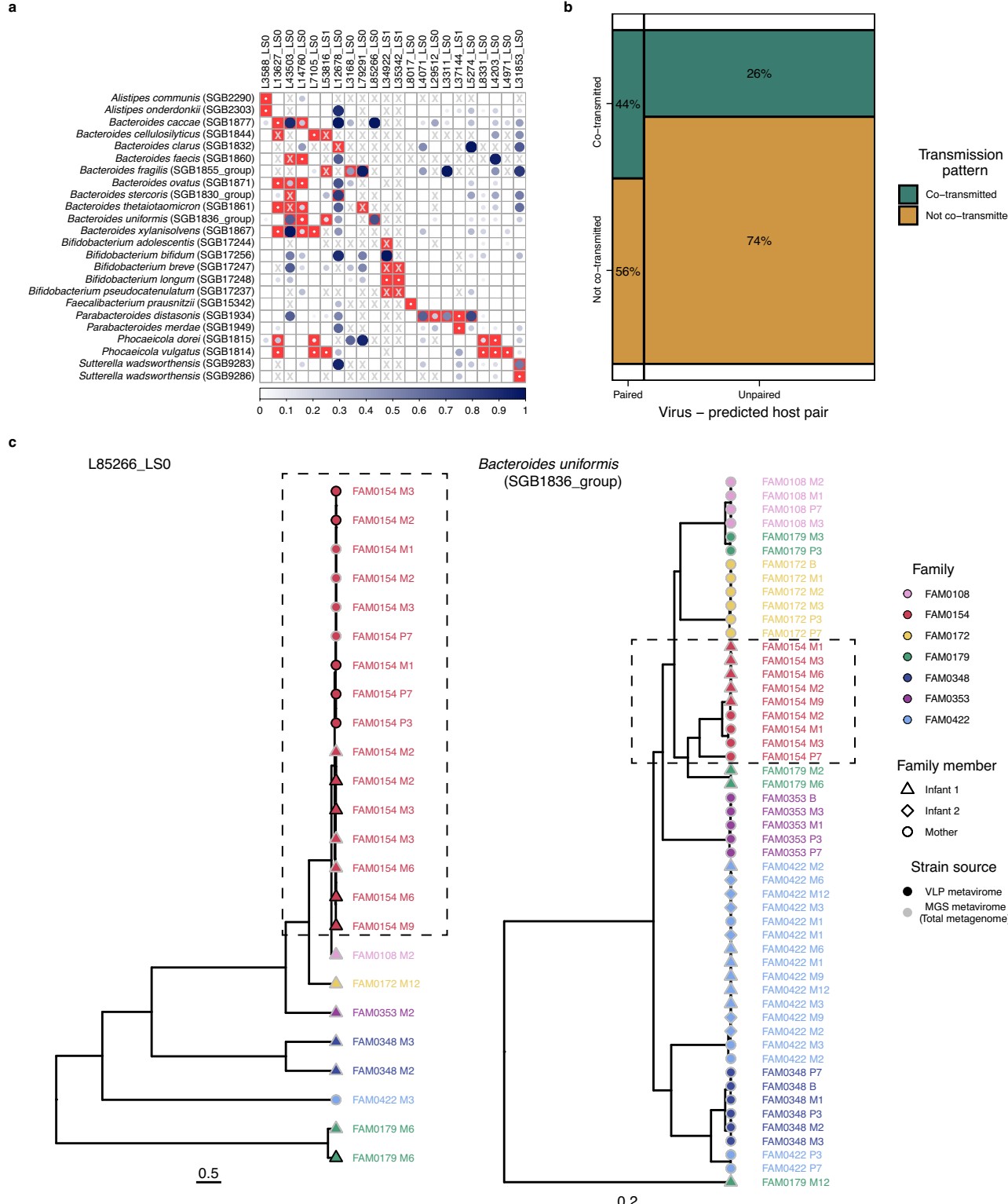

**Fig. 5 | Co-transmission of viral strains and their bacterial host strains from maternal to infant guts. a** Correlation matrix showing the Pearson correlation coefficient for the co-occurrence between viruses and their bacterial host transmission events in concurrent VLP or MGS metaviromes and total metagenomes of mother-infant pairs. X- and Y-axes list viruses and bacterial strains shared within mother-infant pairs, respectively. Red squares at the intersections within the correlation matrix denote the predicted pairs of viruses and their bacterial hosts. Circle size and colour intensity indicate the correlation's strength. A square with an X indicates cases in which it was not possible to derive correlation coefficient for bacterium-virus pair co-transmission (Methods). Blank squares and white circles in red squares indicate the FDR-insignificant correlations (> 0.1). **b** Mosaic plot of the proportions of viruses co-transmitted (in green) and not co-transmitted (yellow) with their predicted bacterial hosts. **c** Phylogenetic trees of a virus (L85266_LS0) and its predicted host (*Bacteroides uniformis* SGB1836_group). Colours indicate the mother-infant pair (family) affiliation. Shape indicates infant or mother sample. Symbol border indicates whether the strain was reconstructed in a total metagenome/MGS metavirome (light grey) or VLP metavirome (black). Dashed rectangles indicate the example of similar topology between phylogenetic trees of a virus and bacterial strain. Source data are provided as a Source Data file.

tree (Fig. 5c). We also found evidence for non-random co-occurrence of phages and their bacterial host in 14 out of 32 virus–host pairs (non-random linkage FDR < 0.05; Methods; Supplementary Data 41), indicating a common co-transmission mechanism throughout different mother-infant pairs. This co-occurrence was more often seen between phage–host pairs than between phage–unrelated bacterium pairs (Fisher test, $p$ value = 0.02). This co-transmission was observed for multiple species of the genus *Bacteroides* (7), *Bifidobacterium bifidum* and *Sutterella wadsworthensis*.

### Possible mechanisms for the origin of the early-life virome
Having established that there are cases of viral strain transmission from mother to infant, we sought to explore the mechanisms underlying the colonisation of bacteriophages in the infant gut. Among the 26 transmitted viruses, 21 were identified as virulent bacteriophages. Based on their predicted lytic lifecycle, it is likely that they were directly seeded in the form of virus particles from the maternal to infant gut in contrast to co-transmission as prophages within the transmitted bacterial strains.

We next focused on investigating the origin of temperate phages in the infant gut. One of the strongest cases of co-transmission between bacteriophages and bacteria was *B. bifidum* and its predicted temperate bacteriophage L34922_LS1 (r = 1, FDR = 0.04; Fig. 5a; Supplementary Data 40). The phylogenetic trees of this temperate phage and its bacterial host are topologically very similar (Fig. 6a). We postulated that the high co-transmission rate we observed might be attributed to the temperate nature of L34922_LS1, which enables it to integrate and be transmitted within the genome of its host. As we reconstructed L34922_LS1 in VLP metaviromes (Fig. 6b), we wondered whether this was coming from phage induction from its transmitted bacterial host. To investigate this hypothesis, we initially reconstructed the genome of *B. bifidum* from metagenomes in which the presence of L34922_LS1 was detected. Next, we mapped the L34922_LS1 genome sequence to the genome of *B. bifidum* and observed a high identity (>99%) and coverage (100%) for the L34922_LS1 sequence (Fig. 6c), which confirmed that this phage could be observed in the prophage form within the *B. bifidum* genome. Additionally, we detected both the integrase and viral recombination genes in the L34922_LS1 sequence (Fig. 6d), further confirming the phage's ability to integrate into the host genome and undergo recombination. Metagenomic read alignment to the genome of *B. bifidum* containing L34922_LS1 revealed that 96.2% of *B. bifidum* strains do not contain L34922_LS1 at the determined region of prophage insertion (Fig. 6e). However, in *B. bifidum* genomes detected in samples from mother-infant pairs where L34922_LS1 transmission was observed, the prophage insertion region was sufficiently covered by reads. VLP metaviromic read-alignment profiles demonstrated a consistent increase in read coverage at the region of prophage insertion compared to the rest of the *B. bifidum* genome, confirming prophage induction in infant and maternal samples (Fig. 6f). Overall, these observations suggest that L34922_LS1 observed in the infant gut originated from the *B. bifidum* of the mother.

We next attempted to find the origin of temperate bacteriophages that were not shown to be significantly transmitted from mother to infant. One of these was a temperate bacteriophage identified in infant samples, L37775_LS1, that is predicted to infect multiple species of the *Bifidobacterium* genus. After mapping the genome sequence of L37775_LS1 to patched *Bifidobacterium* genomes reconstructed from total metagenomes concurrent to VLP metaviromes carrying L37775_LS1, we narrowed down the host range to *Bifidobacterium scardovii* (identity 100%, coverage 100%), which was absent in maternal samples. MGS and VLP metaviromic read-alignment profiles to the *B. scardovii* genome revealed that L37775_LS1 is present at the indicated prophage region and can be induced from its host (Supplementary Fig. 6a–c). As *B. scardovii* genome was only present in infants,

our observations suggest that its phage L37775_LS1 does not originate from the maternal gut. Gene annotation showed that, in addition to carrying an integrase gene, L37775_LS1 also carries a CAZyme gene (Glycosyl hydrolases family 25, Supplementary Fig. 6d).

## Discussion
In this study, we characterised the faecal bacteriome and virome in 30 mothers and their 32 infants longitudinally during pregnancy, at birth and during the first year of life. To our knowledge, this is one of the few studies to look at the maternal virome longitudinally during pregnancy, birth and after birth. In the maternal bacteriome, we observed a notable shift in composition between the first and second trimesters of pregnancy. This is in line with the findings of Koren et al., who proposed that hormonal shifts, immune system adaptations and dietary variations during pregnancy can impact the composition of gut bacteria[26,27]. During late pregnancy, birth and after birth, however, we observed that the overall composition of the maternal gut bacteriome and virome do not change. These results suggest that once established during the second trimester of pregnancy, the maternal gut bacteriome and virome remain consistent throughout this critical period of maternal and infant health.

We demonstrated that the infant gut virome during the first year of life was highly dynamic, and while it progressively transitioned to resemble an adult-like virome with time, it was still very different from that of the mother at the age of one year. We show that the infant gut has a high proportion of temperate phages in the first 3 months of life and that this proportion decreases drastically at 6 months. We thus hypothesise that temperate phages are fundamental in seeding the gut virome, most likely through prophage induction of pioneering gut bacteria. However, even at the age of one year, the abundance of temperate phages still remained higher than that observed in adults, indicating ongoing viral development. Altogether, our results indicate there is a high degree of prophage induction in the gut during the first 3 months of infancy that is followed by stabilisation of the gut environment at a later age as the availability of more bacterial hosts allows for more prophage integration.

Our results highlight the influence of infant feeding mode on infant virome composition. The higher alpha diversity we observed in exclusively formula-fed infants may be attributed to the different composition of formula compared to breast milk. The increased richness of active temperate bacteriophages in exclusively formula-fed infants suggests that formula feeding may provide specific nutrients or environmental conditions that promote the proliferation of temperate phages. The consistent association observed for both active temperate phages and prophages supports the notion that formula feeding has a lasting impact on the acquisition and maintenance of temperate phages in the infant gut. Our investigation into vOTUs grouped by their host bacteria reveals an intriguing finding related to *Bacteroides fragilis*. Although the relative abundance of *B. fragilis* itself was similar between feeding groups, formula feeding was associated with a higher presence of active temperate phages specifically targeting *B. fragilis*. This suggests that formula feeding may induce the production of bacteriophages that target this bacterial species, independent of changes in bacterial abundances. The mechanisms underlying this association warrant further investigation.

Only a few studies have addressed infant gut viromes in relation to maternal viromes[7,21,22]. Duranti et al. focused entirely on the transmission of *Bifidobacterium* phages from mother to infant gut and showed that these could be transmitted[7]. Our study showed that not only *Bifidobacterium* phages but numerous phages predicted to infect bacteria from other genera like *Bacteroides* were also transmitted from mother to infant. A recent study by Walters et al. in 53 infants showed that the overall infant gut virome composition was not driven by exposure to mothers but rather by dietary, environmental and infectious factors[22]. However, in this study, no direct comparison was made

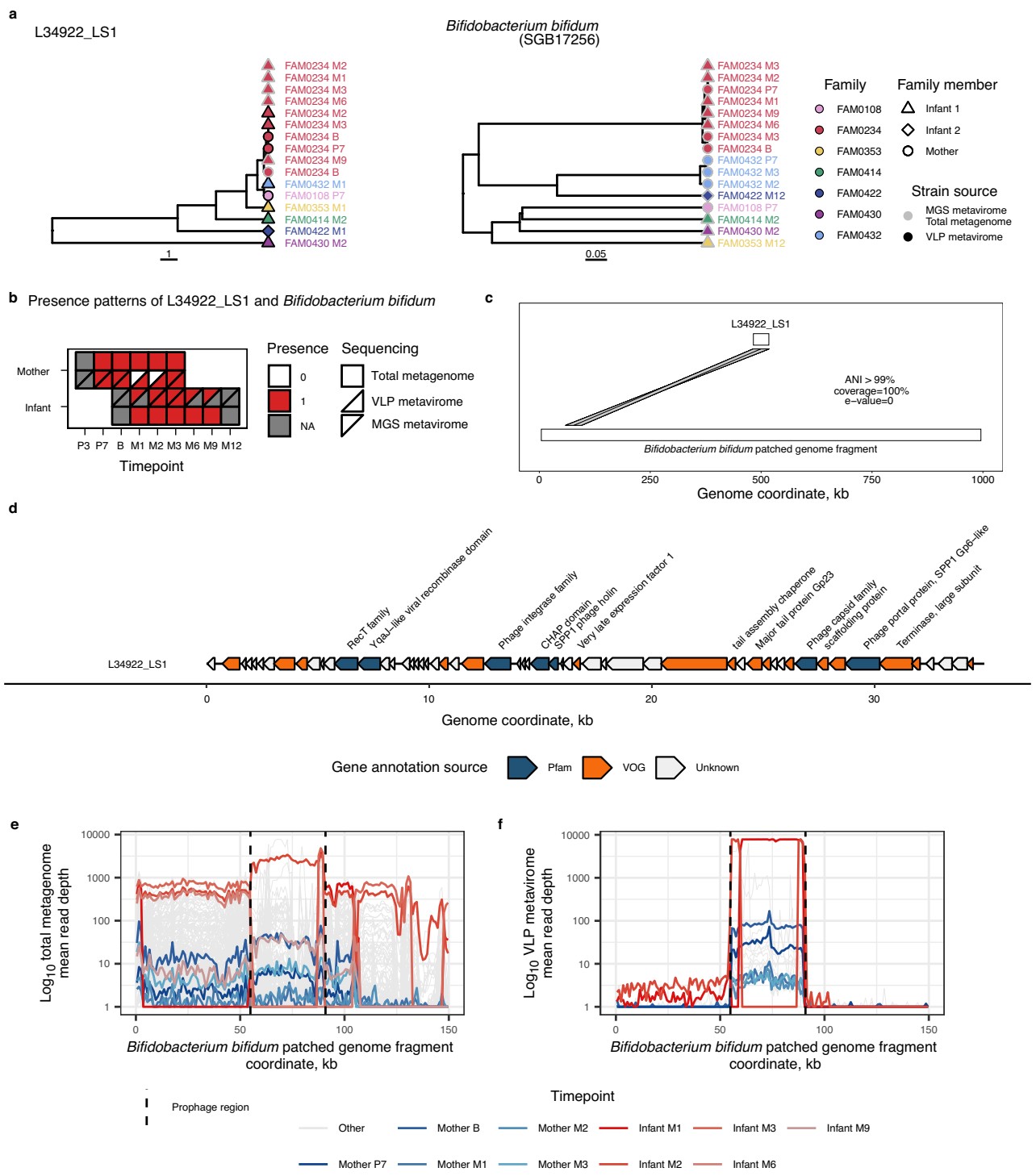

between infant and maternal gut microbial strains. Another study by Maqsood et al. comparing virus scaffold presence-absence between mother and infant in 28 infant twin pairs showed that, on average, 15% of the infant virome was shared with their own mother's gut virome[21]. Our study revealed that, despite significant distinctions between the infant and maternal gut viromes, infants shared on average 32.7% of vOTUs with their mothers. This difference in sharedness may be attributed to the fact that our study encompassed longitudinal samples from both mother and infant for a longer timeframe, whereas Maqsood et al. looked at the sharing of vOTUs between mother and infant cross-sectionally around birth. However, to make definitive claims about transmission, one cannot merely rely on virus sequence co-occurrence in maternal and infant viromes, and it is essential to examine the genetic makeup of viral strains, their sequence similarity and to define a strict strain identity discrimination threshold, an aspect not explored in the above-mentioned studies. For vOTUs shared between the infant and maternal guts, we constructed strains and showed that more than half of the strains were shared more frequently within related mother-infant pairs compared to unrelated mother-infant pairs. This shows that, while it might not be the most defining factor of the infant gut virome, infants do share viral strains with their mother's gut. In microbiome transmission studies, determining the

**Fig. 6 | Example of co-transmission of a temperate phage and its bacterial host: a temperate phage of *Bifidobacterium bifidum*. a** Phylogenetic trees of a temperate phage (L34922_LS1) and its predicted host (*Bifidobacterium bifidum* SGB17256). Colours indicate the mother-infant pairs (family) affiliation, and shapes indicate infant or mother sample. Symbol borders show whether the strain was reconstructed from a total metagenome/MGS metavirome (light grey) or VLP metavirome (black). **b** Presence patterns of L34922_LS1 and *B. bifidum* in the samples of the mother-infant pair (FAM0234) where the transmission of L34922_LS1 was detected. X-axis indicates timepoints, Y-axis shows the mother or infant sample. Shape indicates total metagenome (square), VLP metavirome (triangle) or MGS metavirome (rotated triangle). Colour of the shape indicates the presence (red) or absence (white) of the virus or bacterial strain in the sample, and the missing samples are coloured in grey. **c** Synteny plot of L34922_LS1 genome sequence mapping to the fragment of the *B. bifidum* genome fragment. X-axis indicates the genome coordinates (in kilobases). Lines connecting the L34922_LS1

and *B. bifidum* genome fragment indicate the prophage insertion region. **d** Genome organisation of L34922_LS1. X-axis depicts the genome coordinates in kilobases. A polygon represents every predicted protein, and its orientation indicates the location of the predicted protein at the positive (right-orientation) or negative (left-orientation) strands. Colours indicate the source of protein annotation: Pfam (blue), VOG (orange). Proteins with no functional annotations are shown in white. Orange and blue polygons without text indicate hypothetical proteins with unknown function. **e** Depth of the *B. bifidum* genome fragment coverage by sequencing reads from total metagenomes that were positive for *B. bifidum*, and (**f**) VLP metaviromes that were positive for L34922_LS1. Each transparent grey line corresponds to a sample and represents mean depth in a 1,001-nt sliding window. X-axis depicts the genome coordinate in kilobases. Coloured lines represent samples from the mother-infant pair (FAM0234), and colours show timepoints. Dashed lines indicate the prophage insertion region. Source data are provided as a Source Data file.

directionality of strain inheritance poses a fundamental challenge. Although it is theoretically possible for an infant to acquire a strain and transmit it to the mother, this scenario is improbable given the greater diversity and stability of the adult gut microbiota. The acquisition of a shared strain by both individuals from a common environmental source is another plausible explanation. However, the prevailing consensus in the field suggests that the primary direction of transmission is from mother to infant[12] which is what we assume in this paper. Our findings suggest that some viral and bacterial strains are co-transmitted between related mother-infant pairs, and this holds especially true for species of the genus *Bacteroides, Bifidobacterium bifidum* and *Suttertella wadsworthensis*. We then show examples of how this might occur via direct transmission of bacteriophages and prophage induction following transmission of their bacterial hosts. We also show how infants obtain some viral strains from induction from bacteria that did not come from their mothers.

Our study has several strengths. Firstly, our quantitative approach (avoidance of amplification techniques during the isolation and sequencing of VLP DNA) allowed accurate quantification of viruses and led to minimal bias in our estimation and characterisation of dsDNA viruses. Furthermore, by utilising both total and viral-enriched metagenomes, we could characterise the whole virome, including prophages. Additionally, our dense longitudinal sampling design in both mothers and infants allowed us to study viral compositional dynamics in critical periods such as pregnancy, birth and post-birth. Our study also addressed previously unstudied factors for maternal-to-infant gut microbial transmission such as the place of delivery. As home deliveries constitute 12.7% of the total deliveries (2018) in the Netherlands and 28% in our samples, we had a unique opportunity to explore the effect of place of birth on the viral and bacterial communities of the gut, although we note that our sample size is small and larger studies are needed to make strong conclusions about this.

Our study has several limitations. Firstly, due to our isolation method, we focused solely on the dsDNA viruses present in the gut and overlooked the single-stranded DNA viruses and RNA viruses. While RNA viruses are typically perceived to have a lower abundance in the healthy human gut, it is crucial for future investigations to delve into this under-studied aspect of the infant virome. Including single-stranded DNA viruses and RNA viruses in future research will shed light on their potential roles within the gut ecosystem. Our sample size was also small, which hampers associations with phenotypes and limits the generalisability of the findings to the whole population. Due to the small number of participants who had C-sections, we could not study the effect of this important factor on the gut virome even though an impact of birth mode on the early-life gut virome has previously been described[30]. Thirdly, while we use the term "human gut virome" throughout this study to be consistent with earlier studies, we acknowledge that this result may be biased as faecal samples do not accurately take into account the viruses residing in the gut mucosa.

Hence our results regarding the gut virome are limited to faecal viromes. Finally, despite our best efforts, we cannot guarantee that the viral scaffold database is free from bacterial contamination.

In conclusion, we characterised the gut bacteriome and virome in 30 mothers and their 32 infants from birth to the first year of life using a complementary approach examining both total metagenomes and viral metagenomes. We show that the maternal virome composition does not change significantly during late pregnancy, birth and after birth, whereas the infant gut virome composition is highly dynamic in the first year of life and is influenced by the infant's feeding mode and place of delivery. Initially, infants' viromes have a high proportion of active temperate bacteriophages, which decrease over time but remain higher than in adults at one year of life. Lastly, we provide evidence of viral and bacterial strain co-transmission from mothers to infants, indicating that infants acquire some of their virome from their mothers. Moving forward, future investigations should focus on elucidating the functional implications of these findings and their potential impact on the long-term health and development of infants.

## Methods
### Compliance with ethical regulations
Our research complies with all relevant ethical regulations. The samples utilised in this study were obtained from the Lifelines NEXT cohort study[31,32], that was approved by the Ethics Committee of the University Medical Center Groningen (document number METC UMCG METc2015/600). Written informed consent forms were signed by the participants or their parents/legal guardians, granting permission for further use of clinical data and biological material for research purposes.

### Study cohort
The mother-infant pairs that collected their samples for this study were a part of Lifelines NEXT cohort[31], a birth cohort designed to study the effects of intrinsic and extrinsic determinants on health and disease in a four-generation design[32]. Lifelines NEXT is embedded within the Lifelines cohort study, a prospective three-generation population-based cohort study recording the health and health-related aspects of 167,729 individuals living in the Northern Netherlands[33]. From 2016 to 2023, we included 1450 pregnant Lifelines participants in Lifelines NEXT and intensively followed them, their partners and their children up to at least 1 year after birth. During the Lifelines NEXT study, biomaterials, including maternal and neonatal (cord) blood, placental tissue, faeces, breast milk, nasal swabs and urine are collected from the mother and child at ten timepoints. Furthermore, data on medical, social, lifestyle and environmental factors are collected via questionnaires at 14 different timepoints and via connected devices. Participants of the present study were enroled from 2016 to 2017. Exact age was collected and used as a covariate for relevant analysis. In addition to this, information regarding gestational age, infant sex,

birth weight, feeding mode, mode & place of birth was collected via questionnaires and prioritised for analysis.

## Sample collection

Mothers collected their faeces during pregnancy at weeks 12 and 28, very close to birth, and during the first 3 months after birth (Fig. 1a). Faeces from infants were collected from diapers by their parents at 1, 2, 3, 6, 9, and 12 months of infant age. Parents were asked to use stool collection kits provided by the University Medical Center Groningen (UMCG) and freeze the stool samples at home at −20 °C within 10 min of stool production. Frozen samples were then collected by UMCG personnel and transported to the UMCG in portable freezers and stored in a −80 °C freezer until extraction of microbial and virus-like particle (VLP) DNA. For this study, 361 faecal samples were collected.

## Microbial DNA extraction from faecal samples

Total microbial DNA was isolated from 0.2–0.5 g faecal material using the QIAamp Fast DNA Stool Mini Kit (Qiagen, Germany) using the QIAcube (Qiagen) according to the manufacturer's instructions, with a final elution volume of 100 μl. Two negative controls consisting of Milli-Q water were also processed using the same kit and procedure as for faecal samples. DNA eluates were stored at −20 °C until further processing.

## VLP DNA extraction from faecal samples

Out of 361 faecal samples, 259 were selected for VLP enrichment and VLP DNA extraction based on the amount of faecal material collected. This included maternal samples from 28 weeks of pregnancy, birth, and months 1, 2 and 3 after delivery and infant samples at birth and months 1, 2, 3, 6, and 12 after birth (Fig. 1a). To study the gut virome, DNA was extracted from VLPs as described previously[34]. Briefly, 0.5 g faecal material was resuspended in a 10 ml SM buffer (50 ml 1 M UltraPure™ 1 M Tris-HCI Buffer, pH 7.5 (Invitrogen™ #15567027); 20 ml 5 M NaCl (Sigma-Aldrich Cat#221465); 8.5 ml 1 M MgSO₄ (Sigma-Aldrich Cat#230391); 921.5 ml H₂O), then centrifuged at 4800 g for 10 min at 4 °C, followed by supernatant collection and repeated centrifugation. The supernatant was filtered twice through a 0.45-μm pore polyethersulfone membrane filter to obtain the VLPs. The VLPs were concentrated with Polyethylene glycol 8000 (Sigma-Aldrich, Cat#P2139), overnight precipitation and purification by chloroform treatment. The resulting fraction was treated with 8 U of TURBO DNase (Ambion/Thermo Fisher Scientific Cat#AM2238) and 20 U of RNase I (Thermo Fisher Scientific Cat#10568930) at 37 °C for 1 h before inactivating enzymes at 70 °C for 10 min. Subsequently, proteinase K (40 μg, Sigma-Aldrich, Cat#2308) and 20 μl of 10% SDS were added to the samples and incubated for 20 min at 56 °C. Finally, VLPs were lysed by the addition of 100 μl of Phage Lysis Buffer (4.5 M guanidinium isothiocyanate (Sigma-Aldrich Cat#50983), 44 mM sodium citrate pH 7.0 (Sigma-Aldrich Cat#C8532), 0.88% sarkosyl (Sigma-Aldrich Cat#5125), 0.72% 2-mercaptoethanol (Sigma-Aldrich Cat#M6250) and incubated at 65 °C for 10 min. Nucleic acids were extracted twice from lysates using Phenol/Chloroform/Isoamyl Alcohol 25:24:1 (Thermo-Fisher Scientific Cat#10308293) treatment followed by centrifugation at 8000 g for 5 min at room temperature. The resulting aqueous phase was subjected to the final round of purification using the DNeasy Blood & Tissue Kit (Qiagen Cat#69506) with a final elution volume of 50 μl. For the negative controls, four samples of SM buffer alone were run through the VLP enrichment and DNA extraction process. The resulting VLP DNA eluates were stored at −20 °C until further processing.

## Genomic library preparation and sequencing

Faecal microbial DNA and VLP DNA samples were sent to Novogene, China, for genomic library preparation and shotgun metagenomic sequencing. Sequencing libraries were prepared using the NEBNext® Ultra™ DNA Library Prep Kit or the NEBNext® Ultra™ II DNA Library

Prep Kit, depending on the sample DNA concentration, and sequenced using HiSeq 2000 sequencing platform with 2 × 150 bp paired-end chemistry (Illumina®). On average, 30.2 ± 5.0 million paired-end total metagenome reads and 27.4 ± 6.9 million paired-end VLP metavirome reads were generated per sample. Of 361 samples prepared for total microbiome analysis, 326 were successfully sequenced, and 86% (n = 30) of the failed samples were extracted from meconium. The total metagenome negative controls also failed library preparation and, therefore, could not be sequenced. Following this, low read-depth samples (<5 million reads) and one mislabelled sample were excluded, leaving 322 total metagenome samples for analysis.

For virome analysis, 205 of 255 samples were successfully sequenced. As with the total microbiome, the meconium samples from the VLP DNA isolation protocol failed sequencing. Three out of four negative VLP metavirome controls failed library preparation and could not be sequenced. The only successfully sequenced negative VLP metavirome control had 24.4 million paired-end reads, had a high genomic bacteria DNA contamination (36.5%, see below) and consisted mainly of Sphingomonas spp. and Human adenovirus C. This negative VLP metavirome control was used to remove contaminating sequences from the rest of VLP metaviromes (see below in "Profiling of gut virome composition").

## Profiling of total gut microbiome composition

Total metagenome sequencing reads were quality-trimmed, and Illumina adaptor sequences were removed using BBMap (v38.98)[35] and KneadData tools (v0.10.0)[36], resulting in an average PHRED quality score of 33. Following trimming, the KneadData-integrated Bowtie2 tool (v2.4.2)[37] was used to remove reads that aligned to the human genome (GRCh38.p13), and the quality of the processed data was examined using the FastQC toolkit (v0.11.9)[38]. Taxonomic composition of total metagenomes was profiled using the MetaPhlAn4 tool with the MetaPhlAn database of marker genes mpa_vJan21 and the ChocoPhlAn SGB database (202103)[39].

## Profiling gut virome composition

VLP metavirome sequencing reads underwent quality trimming and human reads removal, as described above. Bacterial contamination of VLP metaviromes was assessed by aligning reads to the single copy chaperonin gene cpn60 database[40]. On average, VLP metaviromes contained 8.3% (95% CI: [7.2; 9.6]) of bacterial genomic DNA per sample.

We used a de novo assembly approach to annotate the composition of the gut virome. Specifically, SPAdes (v3.14.1)[41] was utilised in metagenomic mode ('-meta') with default settings to perform de novo assembly per VLP metavirome. The average number of assembled scaffolds was 283,893 for maternal samples and 103,192 for infant samples. Scaffolds smaller than 1 kbp were removed. Scaffolds that were at least 1 kbp underwent rigorous filtering per sample for the following gut virome annotation. The Open Reading Frames (ORFs) in these scaffolds were predicted using Prodigal v2.6.3[42] in metagenomic mode. Ribosomal proteins were identified using a BLASTp[43] search (e value threshold of $10^{-10}$) against a subset of ribosomal protein sequences from the COG database (release 2020). We used a Hidden Markov Model (HMM) algorithm (hmmsearch from HMMER v3.3.2 package)[44] to compare amino acid sequences of predicted protein products against the HMM database Prokaryotic Virus Orthologous Groups (pVOGs)[45]. Hits were considered significant at an e value threshold of $10^{-5}$. To detect viral sequences, VirSorter v1.0.3[46] was run with its expanded built-in database of viral sequences ('−db 2' parameter) in the decontamination mode ('--virome'). Scaffolds larger than 1 kbp were considered putatively viral if they fulfilled at least one of six criteria, similar to those described previously: (1) BLASTn alignments to a viral section of NCBI RefSeq (release 211) with e value ≤ $10^{-10}$, covering >90% of sequence length at

>50% Average Nucleotide Identity (ANI), (2) having at least three ORFs, producing HMM-hits to the pVOG database with an *e* value ≤ $10^{-5}$, with at least two ORFs per 10 kb of scaffold length, (3) being VirSorter-positive (all six categories, including suggestive), (4) being circular[47], (5) BLASTn alignments (*e* value ≤ $10^{-10}$, >90% query coverage, >50% ANI) to 1489 *Crassvirales* dereplicated sequences (99% ANI and 85% length) larger than 50 kbp from the NCBI database (taxid:1978007) and published datasets[48–51] and (6) being longer than 3 kbp with no hits (alignments >100 nucleotides, 90% ANI, *e* value of $10^{-10}$) to the nt database (release 249). 281,789 scaffolds fulfilled at least one of these six criteria.

To remove putative cellular contamination from the virus sequences, 281,789 putative virus scaffolds were dereplicated at 99% ANI with all assembled negative control scaffolds with no filtration applied other than the size of the scaffold (larger than 1 kbp) using CheckV at 85% alignment fraction (relative to the shorter sequence)[52]. Sequence clusters containing negative control scaffolds were excluded from further consideration. The remaining 280,633 putative virus scaffolds were dereplicated at 95% ANI and 85% length to represent vOTUs at the species level[25]. The resulting 110,526 vOTU representatives were screened for the presence of ribosomal RNA (rRNA) genes using a BLASTn search in the SILVA 138.1 NR99 rRNA genes database[53] with an *e* value threshold of $10^{-3}$. An rRNA gene was considered detected in a scaffold if the gene and the scaffold produced a hit covering >50% of the gene length. Additionally, vOTU representatives were clustered with 1489 dereplicated *Crassvirales* sequences larger than 50 kbp and the genomes of the reference database "ProkaryoticViralRefSeq211-Merged" using vConTACT2 v0.11.3 with default parameters[54]. Sequences assigned the status 'Overlap', 'Singleton' and 'Outlier' by vConTACT2 were treated as genus viral clusters consisting of a single scaffold. The resulting viral clusters (VCs) of putative viral scaffolds were subjected to a second decontamination procedure based on the following criteria. VCs were excluded if any of the cluster members: (1) contained an rRNA gene, (2) contained ≥1 ribosomal protein gene and <3 pVOGs per 10 kb, was VirSorter-negative and non-circular and (3) contained >3 ribosomal protein genes. Specific members of VCs were retained if they satisfied any of the following criteria: (1) circular and had ≥1 pVOGs, (2) circular and VirSorter-positive or (3) VirSorter-positive and had no ribosomal protein genes. The final curated database of virus sequences generated from our dataset included 102,280 vOTU representatives ranging in size from 1 kbp to 476 kbp.

To align quality-trimmed VLP metavirome reads to the final curated vOTU representatives, we used Bowtie2 v2.4.5 in 'end-to-end' mode. A count table was then generated using SAMTools v1.14[55]. The sequence coverage breadth per scaffold was calculated per sample using the SAMTools v1.14 'mpileup' command. To remove spurious Bowtie2 alignments, read counts with a breadth of sequence coverage less than 1 × 75% of a scaffold length were set to zero[37]. Consequently, 102,210 virus sequences were utilised for the construction of the final count table, recruiting on average 89.3% of quality-trimmed reads per VLP metavirome sample. The metadata for these 102,210 virus sequences can be found in Supplementary Data 42, including the indications based on inclusion and exclusion criteria for the virus scaffolds. A reads per kilobase per million reads mapped (RPKM) value transformation was applied to the final count table, which was then used for downstream analyses.

Temperate bacteriophages were identified using either the presence of integrase genes or the co-presence of recombinase genes with CI-repressor-like protein genes from the pVOGs annotation within a vOTU representative. The complete list of pVOGs used for temperate phage assignment is available at https://github.com/GRONINGEN-MICROBIOME-CENTRE/LLNEXT_pilot/blob/main/Virome_discovery/temperate_pVOGs_uniq.txt.

## Generation of MGS metaviromes

Quality-trimmed reads from 322 total metagenomes were aligned to the curated virus database on a per sample basis using Bowtie2 v2.4.5 in 'end-to-end' mode[37]. The average alignment rate from total metagenomes to the curated virus database was 50.6 ± 18.2%. As for VLP metaviromes, a count table was generated and transformed for MGS metaviromes. For samples where both VLP and MGS metaviromes were available (204 out of 322), the median Bray-Curtis distance between VLP and MGS metaviromes of the same faecal sample comprised 0.86 ± 0.20. This distance to the own concurrent sample was lower than to samples of unrelated individuals that averaged at 0.99 ± 0.05 (*p* value < 0.001, one-sided Wilcoxon rank sum test run through 1000 permutations).

## Prediction of viral hosts

Virus host assignment was performed using the iPHoP (v1.2.0) framework with the default settings (FDR < 10%) and the database "Sept_2021_pub"[29]. In total, the microbial host was predicted for 68,299 of 102,210 viral scaffolds (66.8%) at the genus level and for 85,135 (83.3%) of all vOTUs at the species level. In cases where multiple hosts were predicted for a virus sequence at the genus or species level, we selected the host taxonomy with the highest Confidence.score from iPHoP. To ensure consistency with the bacterial taxonomic annotation of MetaPhlAn4, the predicted host taxonomy from iPHoP was manually curated. For associations with phenotypes, the RPKM counts of vOTUs were aggregated based on the genus and species levels of the predicted host taxonomy.

## Ecological measurements and statistical analyses

To assess bacterial and viral alpha diversity, no filters were applied to the relative abundance (bacteria) and RPKM counts (viruses). The alpha diversity for both the bacteriome and virome was calculated as the Shannon diversity index using the *diversity()* function in R package *vegan* v.2.6-4[56].

Beta diversity analysis of the virome and microbiome communities was performed at the vOTU and bacterial species levels using Bray-Curtis dissimilarity. The Bray-Curtis dissimilarity between samples was calculated using the function *vegdist()* from the R package *vegan*. We used nonmetric multidimensional scaling (NMDS) to visualise the similarity of bacteriome and virome samples. For that, the function *metaMDS()* from the R package *vegan* was employed with 2 dimensions for visualisation purposes and 1 dimension for the analyses related to biome composition changes. Additionally, *envfit()* with 999 permutations was used to determine the correlation between NMDS and timepoint along with the vector coordinates for Fig. 1b-e.

To test the difference in overall composition of virome and bacteriome (between mothers and infants and between different timepoints), we employed a linear mixed model using *lmerTest* (3.1-3)[57]. The outcome was NMDS1 (dimension). The predictor variable was timepoint (expressed as exact ages in years or days after birth for infants and as continuous timepoint for mothers), and we corrected for the number of quality-trimmed reads and DNA concentration as fixed effects. Individual ID was incorporated as a random effect.

To test the difference in the Shannon diversity index between mothers and infants for bacteriome and virome, we used a linear mixed model. Here the variable tested was sample type (mother or infant), and we corrected for the number of quality-trimmed reads and DNA concentration as fixed effects and considered individual ID as a random effect. A similar linear mixed model was employed to analyse the effect of timepoint on Shannon diversity in mothers and infants separately, with timepoint (expressed as described above) being the predictor variable, and we corrected for the number of quality-trimmed reads and DNA concentration as fixed effects and individual ID as a random effect. Similar linear mixed models were used for the

vOTUs richness (number of detected vOTUs per sample) comparison between mothers and infants.

To compare virome Shannon indices between mothers and infants at 1 year of age, we performed a Wilcoxon rank sum test. To analyse changes in the abundance of vOTUs aggregated at the level of host genus and microbial genus over the first year of an infant's life, a centred log-ratio (clr) transformation was applied using the function *decostand()* from the R package *vegan*. The pseudo count specific to the biome, expressed as half of the minimal abundance in community data, was utilised. Only microbial genera and host genera vOTU aggregates present in more than n (10%) of infant samples were considered. Subsequently, a linear mixed model was used with timepoint (expressed as described above) as the predictor variable and correction for the number of quality-trimmed reads, DNA concentration and mode of delivery as fixed effects and individual ID as a random effect.

We employed a bootstrap resampling approach with replacement to calculate the 95% CIs for the metrics of interest. The goal was to estimate the range within which the true population values for the metrics were likely to fall. We calculated the mean value from each bootstrap sample and repeated this process multiple times ($n = 1000$). A 95% CI was determined by computing the quantiles corresponding to the lower and upper bounds of the distribution (0.025 and 0.975 quantiles, respectively).

### Association of vOTUs aggregated by predicted host and bacterial species with phenotypes

The association analysis with phenotypes was conducted on infant samples using a linear mixed model, focusing exclusively on bacterial species and vOTU aggregates by bacterial hosts present in at least 10% of the infant samples. In each model, we tested the predictor (maternal age, infant sex, feeding mode, birthweight, place of birth and gestational age) as a fixed effect. We further corrected for timepoints (expressed as exact ages in days after birth), the number of quality-trimmed reads, DNA concentration and mode of delivery as fixed effects. Individual ID was included as a random effect.

For all analyses, an FDR correction was applied to correct for multiple testing, with changes considered statistically significant at FDR < 0.05 using the Benjamini-Hochberg method. All statistical tests are two-sided unless explicitly stated otherwise.

### vOTU and bacterial strain-specific analysis

To study viral strain-sharing within mother-infant pairs, we focused on a subset of vOTU representatives that were covered by reads at over 95% of the genome length and shared between maternal and infant VLP and MGS metaviromes. This subset consisted of 4965 vOTUs. We then selected vOTU representatives for further analysis based on the following criteria: 1) a high-quality or complete genome predicted by CheckV or circularised genomes[47,52], 2) sequence length ≥ 3 kbp and 3) present in samples from at least five different families. There were 51 vOTU representatives fulfilling these criteria. For each selected vOTU, we reconstructed consensus sequences for all samples where the vOTU of interest was covered at over 95% of the genome length. We employed the function 'consensus' with flags '-m simple -r' from SAMTools on the read alignments from Bowtie2 output that were used for the RPKM table construction.

We next performed global alignments of consensus sequences per vOTU using *kalign* v1.04[58]. To improve alignment quality, we trimmed 100 bp from both ends of the global alignment, which were enriched in gaps. Pairwise genetic distances were then calculated using the *dist.dna()* function from the R package *ape* v.5.7-1[59] with default parameters resulting in Kimura 2-parameter (pairwise nucleotide substitution rate between strains) pairwise distances. To compare the pairwise Kimura distances for virus strains between samples of related individuals, we used a one-sided Wilcoxon rank sum test with an alternative hypothesis that the distances between strains identified in

samples of related individuals are smaller than the distances between unrelated samples. Significance of the distance comparison was derived in a permutation test with 1000 iterations, designed to account for the highly unequal number of distances between strains of related and unrelated individuals. FDR correction for multiple testing was applied as described above.

To investigate strain-sharing between mothers and infants, we selected those viruses with lower Kimura distances between strains of related individuals compared to unrelated individuals. To define strain-sharing events within mother-infant pairs, we used an approach similar to that used in Valles-Colomer et al[60]. In short, we compared the median-normalised distances within individuals' strains over the entire study period (maximum 9 months for maternal samples and maximum 12 months for infants) to the normalised distances between strains of unrelated individuals, per vOTU. The strain identity cut-off was calculated using the *cutpointr()* function from the R package *cutpointr* v.1.1.2[61]. For the identification of the optimal cutpoint, we used the *oc_youden_kernel* parameter along with the *youden* metric. Additionally, empirical FDR was defined as the 5th percentile of the unrelated individual comparisons when Youden's index was above 5%. We then compared the percentage of shared versus different dominant strains in related and unrelated mother-infant pairs (all timepoint pairs) using the defined strain identity cut-off. If normalised distances between strains were greater than the cut-off, the strains were deemed different. If they were smaller, this was considered a strain-sharing event. This allowed us to calculate a percentage of dominant strain-sharing between related and unrelated mother-infant pairs, which we then tested for significance using the one-sided Fisher's exact test with subsequent FDR correction for multiple testing using Benjamini-Hochberg.

We found 26 viruses to be shared between mothers and infants. Bacterial hosts for 25 of the 26 transmitted viruses were predicted at the species level using iPHoP. All predicted hosts of viruses were used for co-transmission analysis.

### Bacterial-species-specific strain analysis

We reconstructed bacterial strain SNP haplotypes for the predicted hosts of 25 transmitted viruses using StrainPhlAn4[39], resulting in 37 bacterial strain SNP haplotypes. This method is based on reconstructing consensus sequence variants within species-specific marker genes and using them to estimate strain-level phylogenies. We then performed multiple sequence alignment and used the Kimura 2-parameter method from the *EMBOSS* package[62] to create phylogenetic distance matrices that contain the pairwise nucleotide substitution rate between strains. We next employed the same methods for the identification of strain-sharing events as described above.

### Virus–host co-transmission tracking

To determine if the shared viruses were co-transmitted with their predicted bacterial hosts, we employed the Mantel partial test on modified genetic distance matrices for bacterial and virus strains. This test assessed the correlation between the strain-sharing events for bacteria and phages while controlling for longitudinally collected samples.

First, constructed Kimura genetic distance matrices were normalised by the median genetic distance per bacterial strain and vOTU, respectively. Next, the normalised values of genetic distances were replaced with 0 if the distance did not exceed the vOTU- or bacterial-strain-specific cut-off for individual strain variation (Youden index or 5% FDR), otherwise, it was replaced with 1. This modification allowed us to focus on strain-sharing events rather than on the correlation between genetic distances themselves.

Next, for each vOTU and bacterial strain, we selected subsets of concurrent samples where both the vOTU and bacterial strain were reconstructed. For bacterial strains, only total metagenomes were

used. For the viral strains, strains reconstructed from VLP and MGS metaviromes were included. If both types were available for the same individual and timepoint, strains from VLP metavirome samples were prioritised.

To account for repeated measurements, we created control matrices for the selected subsets of concurrent samples. In this matrix, we assigned a value of 0 when the strain was reconstructed in samples from the same individual and a value of 1 when the strain was reconstructed in samples from different individuals. In this analysis, mothers and their infants were treated as different individuals.

The Mantel partial test, using *mantel.partial()* function from the R package *vegan*, was performed on the modified genetic distance matrices for bacteria and viruses. We used the Pearson correlation method and 999 permutations to assess the significance of the correlation while controlling for longitudinal samples using the control matrix. In cases when virus and bacterium distance matrices were collinear, we used *mantel()* function from the R package *vegan*. In cases when one of the distance matrices was collinear with the control distance matrix (e.g., in the subset of concurrent samples, the strain-sharing pattern repeated the structure of the longitudinal samples belonging to the same/different individuals), it was not possible to calculate the Mantel partial test while accounting for the repeated measurements. These pairs of virus-bacterium were not used for the testing of co-transmission frequency. The *p* values obtained from the Mantel partial test were adjusted for multiple testing using FDR correction using the Benjamini-Hochberg method.

Additionally, we conducted a one-sided Fisher's exact test to compare the frequency of co-transmission events between phage-bacteria pairs connected by a virus–host relationship and random phage-bacteria pairs.

To further test the non-randomness of co-sharing patterns of viruses and their bacterial hosts, we calculated the difference between the observed frequency of co-transmission of virus and bacterial strains (i.e. the proportion of concurrent samples where both virus and bacterial strain had a modified genetic distance equal to 0) and the expected frequency based on independent strain sharing events (the product of the fractions of samples with a modified genetic distance of 0 for viruses and bacteria independently). The difference between the observed and expected frequencies is computed and termed "non-random linkage". The significance of non-random linkage was calculated using a chi-squared test.

## Bacterial genome binning and read alignment to patched genomes

Scaffolds were assembled from total metagenomes in the same way as for VLP metaviromes. All scaffolds were then used for genome binning using metaWRAP[63]. Taxonomy was assigned to these bins using GTDB-Tk v2[64]. Bacterial genomic bins were then patched and scaffolded using RagTag[65] with 'scaffold' option and the genomes of isolates from MGnify, if available, or metagenome-assembled genomes[66] (MGYG000132487 for *Bifidobacterium bifidum* and MGYG000003383 for *Bifidobacterium scardovii*). vOTU representatives of interest were then mapped to the patched bacterial genomes using minimap2[67] using '-X -N 50 -p 0.1 -c' flags to locate the prophage region. Next, quality-trimmed reads from total metagenomes and VLP metaviromes were aligned to the patched genomes using Bowtie2 as described above. The sequence coverage breadth per scaffold was calculated per sample using the SAMTools 'mpileup' command. A 1001 nt window with 1001 nt step was used to calculate mean coverage depth; the most 3′-terminal window was extended to include up to 1001 3′-terminal nucleotides.

Gene annotation of vOTU representatives of interest was performed using VIBRANT in '-virome' mode[68].

## Data visualisation

Results were visualised in graphical form using a set of custom R scripts, including calls to functions from the following packages: *ggplot2* v.3.4.2[69], *ggrepel* v.0.9.3[70], *ggforce* v.0.4.1[71], *patchwork* v.1.1.2[72], *tidyverse* v.2.0.0[73], *EnhancedVolcano* v.1.16.0[74], *ggforestplot* v.0.1.0[75], *corrplot* v.0.92[76] and *ggtree* v.3.6.2[77]. All boxplots were prepared using *ggplot2* and represent standard Tukey type with interquartile range (IQR, box), median (bar) and $Q1 - 1.5 \times IQR/Q3 + 1.5 \times IQR$ (whiskers). Phylogenetic trees were built based on the Kimura 2-parameter distance matrices. First, hierarchical clustering was applied to the distance matrices using the function *hclust()* from the R package *stats* v.4.2.1. Clustering trees were then converted into phylogenetic trees with function *as.phylo()* from R package *ape* v.5.7.−1[59]. Visualisation of the phylogenetic trees was done using the function *ggtree()* from the R package *ggtree*[77].

## Statistics & Reproducibility

The current study is a pilot study of the first samples collected in the Lifelines NEXT project, without prior selection. No statistical method was used to predetermine sample size due to the exploratory nature of the study. Of 326 successfully sequenced total metagenomes, three low read-depth samples (<5 million reads) and one mislabelled sample were excluded from the analysis. In the vOTU retention analysis, 11 of 32 infants (with VLP metaviromes at M1) and 26 of 30 mothers (with VLP metaviromes at P7) were included. For the bacterial species retention analysis, 28 infants (with M1 samples) and 20 mothers (with P3 samples) were analysed. PPV analysis included 14 infants (with at least 3 timepoints) and 27 mothers (with at least 3 timepoints). All infants (32) and 27 mothers were included in the PPB analysis due to the availability of the required timepoints. In the analyses concerning feeding practices, 28 infants were included due to the availability of VLP metaviromes and feeding data. In the analyses concerning place of delivery, all infants were included. Our study ensures reproducibility by providing detailed methodological procedures, code, and per-request accessible quality-trimmed human-contaminant-free sequencing data.

## Reporting summary

Further information on research design is available in the Nature Portfolio Reporting Summary linked to this article.

## Data availability

Sample information, basic clinical data, family structure, and quality-trimmed and human-contaminant-free sequencing reads can be found in the European Genome-Phenome Archive (EGA) repository (study ID: EGAS00001005969) and available upon submitting a request using the form provided at https://forms.gle/A4Jem2rMnjcygWRD6. A response will be provided within 15 working days. Access to the Lifelines NEXT Project data will be granted to all qualified researchers and will be governed by the provisions laid out in the LLNEXT Data Access Agreement: https://groningenmicrobiome.org/?page_id=2598. This access procedure is in place to ensure that the data is requested solely for research and scientific purposes, in compliance with the informed consent signed by Lifelines NEXT participants. To assist and facilitate data utilisation from EGA, authors have prepared a comprehensive instruction guide that can be found at: https://github.com/GRONINGEN-MICROBIOME-CENTRE/LLNEXT_pilot/blob/main/Data_Access_EGA.md. The virus scaffolds and their metadata are available in the Figshare repository under https://doi.org/10.6084/m9.figshare.23926593.v2. Source data are provided with this paper. Links to other datasets or databases used in the present study can be found here: Human reference genome GRCh38.p13 [https://www.ncbi.nlm.nih.gov/datasets/genome/GCF_000001405.39/]; MetaPhlAn database of marker genes (mpa_vJan21); https://doi.org/10.1038/s41587-023-01688-w; ChocoPhlAn SGB database (202103); https://doi.org/10.

1038/s41587-023-01688-w; GTDB-Tk, database (release207_v2); https://doi.org/10.1093/bioinformatics/btac672; cpn60 database; https://doi.org/10.1101/gr.2649204; COG database (release 2020); https://doi.org/10.1093/nar/gkaa1018; Prokaryotic Virus orthologous Groups (pVOGs); https://doi.org/10.1093/nar/gkw975; NT database (release 249); https://ftp.ncbi.nlm.nih.gov/blast/db/; iPHoP database (Sept_2021_pub); https://doi.org/10.1371/journal.pbio.3002083; SILVA 138.1 NR99 rRNA genes database; https://doi.org/10.1093/nar/gks1219; ProkaryoticViralRefSeq211-Merged database (release 211); https://ftp.ncbi.nlm.nih.gov/refseq/release/viral/ Source data are provided with this paper.

## Code availability
All codes used in this study can be found at: https://github.com/GRONINGEN-MICROBIOME-CENTRE/LLNEXT_pilot and https://doi.org/10.5281/zenodo.10407473[78].

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

## Acknowledgements

We thank the participants of the Lifelines NEXT cohort for their participation. We thank doctors' assistants and the service bureau team of Lifelines NEXT and acknowledge significant contributions of the persons involved in the collection, processing and storage of samples: Brenda Hijmans, Annet Jansen, Gea Lamberts, Rianne de Roos, Daphne Teuben and Ettje Tigchelaar. We thank Kate Mc Intyre for editing the manuscript. We also thank the Genomics Coordination Center and the Center for Information Technology of the University of Groningen for their support and for providing access to the Gearshift and Peregrine high-performance computing clusters. We thank Daoming Wang for his assistance in the figure designs and Dianne Jansen for her assistance in DNA isolations. This work was supported by scholarships from the Graduate School of Medical Sciences, University of Groningen to SG and the Junior Scientific Masterclass, University of Groningen to TS. SG was awarded a de Cock-Hadders Stitching grant (2021-08). TS was awarded a de Cock-Hadders Stichting grant (Winston Bakker Fonds 2021-WB-08). JES was supported by grants from the de Cock-Hadders Stichting (2021-57) and the International Society for Research in Human Milk and Lactation (ISRHML). SB was supported by EUCAN-connect, a federated FAIR platform enabling large-scale analysis of high-value cohort data connecting Europe and Canada in personalised health. Furthermore, this project was funded by the Netherlands Heart Foundation (IN-CONTROL CVON grant 2018-27 to AZ and JF), the Netherlands Organisation for Scientific Research (NWO) (NWO Gravitation Exposome-NL (024.004.017) to AK and AZ; NWO-VIDI 864.13.013 and NWO-VICI VI.C.202.022 to JF; NWO-VIDI 016.178.056 to AZ and NWO Spinoza Prize SPI 92-266 to CW), the European Research Council (ERC) (ERC Advanced Grant 2012-322698 to CW, ERC Consolidator Grant 101001678 to JF, and ERC Starting Grant 715772 to AZ), the RuG Investment Agenda Grant Personalised Health to CW. JF and CW are also supported by the Netherlands Organ-on-Chip Initiative, an NWO Gravitation project (024.003.001) funded by the Ministry of Education, Culture and Science of the government of the Netherlands. AZ was also supported by EU Horizon Europe Program grant INITIALISE (101094099). CH was supported by Science Foundation Ireland (SFI) under Grant Number SFI/12/RC/2273. ANS was supported by a Wellcome Trust Research Career Development Fellowship (220646/Z/20/Z) and the ERC under the European Union's Horizon 2020 research and innovation programme (ERC Consolidator Grant agreement 101001684). This research was funded in whole, or in part, by the Wellcome Trust [220646/Z/20/Z]. For the purpose of open access, the authors have applied a CC BY public copyright licence to any Author Accepted Manuscript version arising from this submission. The study was also financially supported with a public-private partnership allowance of

Health Holland Topsector Life Sciences & Health to stimulate public-private partnerships. The funders had no role in the study design, data collection and analysis, decision to publish, or preparation of the manuscript.

## Author contributions

S.G., T.S. and A.Z. designed the study. S.G., T.S., J.E.S., S.B., M.K., J.D., J.S., S.S., C.W. and L.L.N. gathered and prepared the data. S.G., T.S., A.G., N.K., and A.K. analyzed the data. AG and NK equally contributed as second authors in this study. S.G. and T.S. wrote the manuscript. S.A.-S. and A.K. provided advice for statistical methods. A.N.S. and C.H. provided advice for metaviromes extraction and annotation. A.G., J.E.S., S.A.-S., R.G., A.V.V., S.B., F.K,. J.F., and A.Z. critically reviewed the manuscript.

## Competing interests

The authors declare no competing interests.

## Additional information

## Lifelines NEXT cohort study

Jackie Dekens[1,5], Aafje Dotinga[10], Sanne Gordijn[9], Soesma Jankipersadsing[1], Ank de Jonge[11,12], Marlou L. A. de Kroon[4], Gerard H. Koppelman[13], Folkert Kuipers ⓘ[2,6], Lilian L. Peters[11,12], Jelmer R. Prins[9], Sijmen A. Reijneveld[4], Sicco Scherjon[9], Jan Sikkema[5], Trishla Sinha ⓘ[1,14], Morris A. Swertz[1], Henkjan J. Verkade[2], Cisca Wijmenga ⓘ[1] & Alexandra Zhernakova ⓘ[1,15]✉

[10]Lifelines Cohort Study, Groningen, the Netherlands. [11]Midwifery Science, Amsterdam University Medical Center, Vrije Universiteit Amsterdam, AVAG, Amsterdam Public Health, Amsterdam, the Netherlands. [12]Department of Primary and Long-term Care, University of Groningen, University Medical Center Groningen, Groningen, the Netherlands. [13]Department of Paediatric Pulmonology and Paediatric Allergology, Beatrix Children's Hospital, University of Groningen, University Medical Center Groningen, Groningen, the Netherlands.

