## [Peer Review File · Nature Communications]

REVIEWER COMMENTS

Reviewer #1 (Remarks to the Author):

Garneva et al present an analysis of a metagenomic analysis of bacteria and dsDNA virus in the stools of ~30 mother and infant pairs followed in a longitudinal fashion. They examine the viral portion in the VLPs and via metagenomic sequencing without viral prep.

They reproduce previous observations on differences in bacterial and viral community composition between mothers and infants, on shared (maternal/infant) bacteria and bacteriophages which is important, but not novel. Relevant papers (not all): Lim et al. 2015, Zhao et al. 2017, McCann et al. 2018, Magsood et al. 2019, Liang et al. 2020, Walters et al 2023.

Major comments:

1. Terms should be defined: metaviromes, whole metavirome, vOTU, metagenome in the beginning- it is confusing if they are referring to viruses detected in VLP prepped sample or virus in the portion of sample that did not get viral prep. This is key throughout and as written is not always clear.
2. Page 11 Line 211. This methodology assumes that any phage detected in the whole metagenome data set are prophage. This does not seem like a valid assumption, as active phage could be captured here as well.
3. Page 16 line 332. What is the rationale behind the conclusion these were through direct seeding.
4. Page 18 line 371. This is quite speculative.
5. Page 19 line 376. This is incorrect. Please see Walters et al Cell Host and Microbe, 2023 (Ref 22).
6. Page 21. Line 421-424. Three papers, Magsood, Walters, and this one, find only a relatively small percent of shared viral component between mother and infant gut. Why invoke that small difference in percent shared is due to longitudinal sampling when the maternal virome is stable over time. Seems that the point is being missed here.
7. Page 21-22. Line 442-444 these conclusions are overstated.
8. Page 27. Line 598. Criteria 6 is concerning. Why are these necessarily thought to be viral? Were possible archaeal and bacterial sequences removed? Did the authors try blastx of these scaffolds? What did they hit by blastx? A table which outlines by which method called a virus would be helpful.
9. Page 28 Line 633. Why did authors choose this method to call temperate phage? Did this agree with other more commonly used tools for this?

10. Page 29. Line 649. It is quite surprising this high level of predicted microbial host. This is not in keeping with other studies using iPHoP.

Minor comments:

1. Page 14 line 269. Please explain what is meant by accounting for prophages detected in whole metaviromes
2. Page 26. Line 556. Seems unusual that 3 out of 4 negative controls failed. Could the authors comment more on this.
3. Page 30 line 686. Is wilcoxon the proper test here?

Reviewer #2 (Remarks to the Author):

The Authors have substantially improved the manuscript by increasing the sampling size and frequency, and by improving the data analysis pipeline. While most of the concerns raised in the first round of Review have been addressed, two relevant issues still persist.

- The sequences and metadata are, differently from what stated by the authors in lines 844-845, not publicly available. The access policy associated to the provided EGA accession says "The Consortium is willing to consider applications from third-party researchers for access to the anonymized sequence data generated by the Project. [...] Access is conditional upon the availability of data and on signed agreement by the researcher to abide by the policies and conditions related to data ownership, disposal, ethical approval, confidentiality and commercialization referred herein". In fact, nor the fastq files nor the metadata can be downloaded. This is unacceptable. This kind of access restriction, entirely omitted in the manuscript, goes against the Journal Guidelines on data availability and against the FAIR data sharing principles.

- The use of "total metagenomes" vs "bacteriome" and "whole metavirome", vs "metavirome" vs "virome" is inconsistent throughout the manuscript and the meaning of these terms is not clearly described, nor in the text nor in the figures.

REVIEWER COMMENTS

Reviewer #1 (Remarks to the Author):

Garneva et al present an analysis of a metagenomic analysis of bacteria and dsDNA virus in the stools of ~30 mother and infant pairs followed in a longitudinal fashion. They examine the viral portion in the VLPs and via metagenomic sequencing without viral prep.

They reproduce previous observations on differences in bacterial and viral community composition between mothers and infants, on shared (maternal/infant) bacteria and bacteriophages which is important, but not novel. Relevant papers (not all): Lim et al. 2015, Zhao et al. 2017, McCann et al. 2018, Magsood et al. 2019, Liang et al. 2020, Walters et al 2023.

Major comments:

1. Terms should be defined: metaviromes, whole metavirome, vOTU, metagenome in the beginning- it is confusing If they are referring to viruses detected in VLP prepped sample or virus in the portion of sample that did not get viral prep. This is key throughout and as written is not always clear.

We agree that the terms, as they were in the manuscript, led to confusion. For the sake of clarity we have now termed metaviromes as either MGS metaviromes or VLP metavirome. Additionally, we have created a box (Box 1) in the introduction as a glossary of the terms used in this paper. Lastly, we have changed the term microbiome to bacteriome in the text wherever applicable. We hope that our modifications increase clarity.

Hereby the text from Box 1:

Bacteriome: community of all bacteria inhabiting an ecosystem

Virome: community of all viruses inhabiting an ecosystem

Total metagenome: the total sum of all genomes of microbes in a sample

VLP metaviromes: the total sum of all genomes of viruses in a VLP-enriched sample

MGS metaviromes: the total sum of all genomes of viruses in a total metagenome

vOTU (virus operational taxonomic unit): species-rank (95% Average Nucleotide Identity, at 85% alignment fraction (relative to the shorter sequence)) virus groups

2. Page 11 Line 211. This methodology assumes that any phage detected in the whole metagenome data set are prophage. This does not seem like a valid assumption, as active phage could be captured here as well.

Original text: Given that the majority of temperate bacteriophages in the adult gut are in the form of prophages²⁷, we compared the percentage of temperate bacteriophages in mothers and infants using the vOTUs detected in whole metaviromes, created by aligning the reads from whole metagenomes aligned to vOTU database reconstructed from metaviromes, thus representing the prophage content of the virome.

We agree with the reviewer that actively replicating temperate phages could be captured by this method. We have therefore rephrased our text. It now reads (lines 207-212):

“Given that the majority of temperate bacteriophages in the adult gut are in the form of prophages²⁷, we compared the percentage of temperate bacteriophages in mothers and infants using the vOTUs detected in MGS metaviromes, created by aligning the reads from total metagenomes aligned to the vOTU database reconstructed from VLP metaviromes, thus representing the prophage-inclusive temperate phage content of the virome.”

3. Page 16 line 332. What is the rationale behind the conclusion these were through direct seeding.

Original text: “Among the 26 transmitted viruses, 21 were identified as virulent bacteriophages. Based on their predicted lifecycle, it is likely that they were transmitted through direct seeding of VLPs from the maternal to infant gut.”

Since one of the criteria for inclusion of vOTUs in the strain-sharing analysis was the genome completeness (high-quality or complete genome predicted by CheckV or circularised genomes), and these 21 mentioned transmitted viruses did not have neither integrase nor site-specific recombinase genes, we concluded that they very likely exclusively undergo lytic infection cycle. For such phages, the only transmission mechanism available is the direct seeding of the viral particles. We agree with the reviewer that this might not have been clear in the original text. Hence, we have rephrased the text (lines 330-333):

“Among the 26 transmitted viruses, 21 were identified as virulent bacteriophages. Based on their predicted lytic lifecycle, it is likely that they were directly seeded in the form of virus particles from the maternal to infant gut instead of being co-transmitted as prophages within the transmitted bacterial strains.”

4. Page 18 line 371. This is quite speculative.

Original text: Gene annotation showed that, in addition to carrying an integrase gene, L37775_LS1 also carries a CAZyme (Glycosyl hydrolases family 25, Supplementary Fig. 6d), indicating that this phage might be associated with infant feeding.

We agree that this is speculative and have adapted the sentence as follows: (lines 368-370): *“Gene annotation showed that, in addition to carrying an integrase gene, L37775_LS1 also carries a CAZyme (Glycosyl hydrolases family 25, Supplementary Fig. 6d).”*

5. Page 19 line 376. This is incorrect. Please see Walters et al Cell Host and Microbe, 2023 (Ref 22).

Original text: To our knowledge, this is the only study to look at the maternal virome longitudinally during pregnancy, birth and after birth. In the maternal total microbiome, we observed a notable shift in composition between the first and second trimesters of pregnancy.

We have now changed this to read (lines 373-375): *“To our knowledge, this is one of the few studies that investigates the maternal virome longitudinally during pregnancy, birth and after birth.”*

6. Page 21. Line 421-424. Three papers, Maqsood, Walters, and this one, find only a relatively small percent of shared viral component between mother and infant gut. Why invoke that small difference in percent shared is due to longitudinal sampling when the maternal virome is stable over time. Seems that the point is being missed here.

Original text: Our study revealed that, despite significant distinctions between the infant and maternal gut viromes, infants shared on average 32.7% of vOTUs with their mothers. This difference in sharedness may be attributed to the fact that our study encompassed longitudinal samples from both mother and infant for a longer timeframe, whereas Maqsood et al. looked at the sharing of vOTUs between mother and infant cross-sectionally around birth.

We thank the reviewer for this comment. We would like to clarify why we believe that our use of longitudinal sampling is leading to the increased sharedness between maternal and infant microbiome:

- 1) Firstly, and most importantly, the study of Maqsood et al. used only one time point of mother and infant. This time was very close to birth. Their study of transmission from mother to infant will thus encompass mostly bacteria and viruses that are vertically transmitted during birth and not arising from the

common shared environment. Hence we expect this percentage to be much smaller than the overall shared microbiome which our study takes into account.

- 2) In pregnancy we indeed show that overall the maternal gut virome remains stable. However, as seen in Figure 2e, there is turnover in viral species which also contributes to the sharedness.
- 3) Finally, having longitudinal sampling gives us a higher chance to detect shared contigs that would be missed when looking simply at one cross-sectional sample.

We have changed the text to clarify this (lines 420-425): *“Our study revealed that, despite significant distinctions between the infant and maternal gut viromes, infants shared on average 32.7% of vOTUs with their mothers. This difference in sharedness may be attributed to the fact that our study encompassed longitudinal samples from both mother and infant for a longer time frame after birth, whereas Maqsood et al. looked at the sharing of vOTUs between mother and infant cross-sectionally around birth which would mostly represent vertical transmission viruses during birth”.*

7. Page 21-22. Line 442-444 these conclusions are overstated.

Original text: We then show examples of how this occurs via direct transmission of bacteriophages and prophage induction following transmission of their bacterial hosts. We also show how infants obtain some viral strains from induction from bacteria that did not come from their mothers.

We disagree with the reviewer here. We specifically mention that we found examples of how this occurs. We have, however, reformulated the text to indicate yet again that this is our hypothesis (lines 442-443): *We then show examples of how this might occur via direct transmission of bacteriophages and prophage induction following transmission of their bacterial hosts. We also show how infants obtain some viral strains from induction from bacteria that did not come from their mothers.*

8. Page 27. Line 598. Criteria 6 is concerning. Why are these necessarily thought to be viral? Were possible archaeal and bacterial sequences removed? Did the authors try blastx of these scaffolds? What did they hit by blastx? A table which outlines by which method called a virus would be helpful.

Original text: (6) being longer than 3 kbp with no hits (alignments >100 nucleotides, 90% ANI, e-value of 10⁻¹⁰) to the nt database (release 249). 281,789 scaffolds fulfilled at least one of these six criteria.

The usage of criteria #6 is in accordance with previously published studies^{1,2}. We agree with the reviewer that criteria #6 alone is not sufficient to call a sequence putatively viral. However, we would like to stress that in our pipeline, we use a robust post-virome discovery filtration process designed using the recent studies in the metaviromics field³⁻⁶. Specifically, we employ screening of putative virus scaffolds for the presence of ribosomal RNA (rRNA) genes using a BLASTn search in the SILVA 138.1 NR99 rRNA genes database along with screening for ribosomal protein genes identified using a BLASTp search (e-value threshold of 10^{-10}) against a subset of ribosomal protein sequences from the COG database (release 2020), as mentioned in the Methods section (lines 609-611 and lines 585-586) and exclude these (lines 619-621). Through this comprehensive approach, we effectively eliminate potential archaeal and bacterial sequences, ensuring that only genuine viral scaffolds are considered in the downstream analysis.

Furthermore, there is an increasing body of evidence^{7,8} that suggests that the NR might contain many prophage proteins that are currently attributed to bacteria as prophages are highly prevalent in gut-associated bacteria. We thus did not blastx these scaffolds to the NR database.

We agree with the reviewer that a table encompassing the indications based on inclusion and exclusion criteria for the virus scaffolds would be helpful. In fact, this table can be found in Figshare along with the fasta file containing all virus scaffolds used in this study:

https://figshare.com/articles/dataset/Virus_scaffolds_reconstructed_in_the_study_b_Mot/23926593

[Infant Gut Viruses and their Bacterial Hosts Transmission Patterns and Dynamics during Pregnancy and Early Life b NEXT Pilot and their metadata /23926593](https://figshare.com/articles/dataset/Virus_scaffolds_reconstructed_in_the_study_b_Mot/23926593)

We have now added this table to the Supplementary tables (S42: Metadata for virus genomes and genome fragments reconstructed in the study).

9. Page 28 Line 633. Why did authors choose this method to call temperate phage? Did this agree with other more commonly used tools for this?

Original text: Temperate bacteriophages were identified using either the presence of integrase genes or the co-presence of recombinase genes with CI-repressor-like protein genes from the pVOGs annotation within a vOTU representative.

The choice for the method to predict the phage lifestyle was based on well-established research in the field (Shkoporov et al.¹ *Cell Host & Microbe*.2019, Howard-Varona et al.⁹*ISME Journal*. 2017, Roux et al.¹⁰ *PeerJ*. 2015). These studies pinpointed integrase and site-specific recombinases as the hallmark genes for the temperate phages. Since site-specific recombinases also regulate viral gene expression during the bacterial cell

infection, our phage lifestyle prediction method included an additional condition of the presence of CI-repressor-like protein genes (Paul.¹¹ *ISME Journal*. 2008). Furthermore, the tools used in the viromics field for the lifestyle prediction like Prophage Hunter, PhiSpy, PHASTEST, BACPHLIP, and PhaTYP also rely on the presence of integrase gene within the virus sequence of interest in the marker-based parts of their pipelines (Song et al.¹² *Nucleic Acids Research*. 2019, Akhter et al.¹³ *Nucleic Acids Research*. 2016, Arndt et al.¹⁴ *Briefings in Bioinformatics*,2019. Hockenberry et al.¹⁵ *PeerJ*. 2019. Shang et al.¹⁶ *Briefings in Bioinformatics*. 2022).

10. Page 29. Line 649. It is quite surprising this high level of predicted microbial host. This is not in keeping with other studies using iPHoP.

Original text: In total, the microbial host was predicted for 68,299 of 102,210 viral scaffolds (67.3%) at the genus level and for 85,135 (82.9%) of all vOTUs at the species level.

Benchmarking of the iPHoP framework in the original article (Roux et al.¹⁷, 2023) showed that the amount of successful host predictions depends on the ecosystem studied. According to the paper, more hosts could be found for predicted viruses in human-associated samples. In our study, the percentage of the viruses with predicted hosts is concordant with the benchmark: “For human-associated microbiomes, about 89% of the nonredundant high-quality genomes had a host predicted using iPHoP, including 57% with very high confidence predictions (iPHoP score ≥ 95)”. (Roux et al. 2023)

To the best of our knowledge, three other studies have employed the iPHoP framework. Specifically, Ter Horst et al.¹⁸ (2023) focused on the virome of wetlands, Peng et al.¹⁹ (2023) on the virome of deep-sea cold seep sediments, and Sáenz et al.²⁰ (2023) on the virome of grass silage. Given that these studies explored non-human associated ecosystems, the proportion of viruses with hosts predicted in their research is anticipated to vary from our findings.

Based on the details previously mentioned, we contend that while the percentage of the predicted host is high, it aligns with the benchmark set by the framework's authors (Roux et al., 2023).

Minor comments:

1. Page 14 line 269. Please explain what is meant by accounting for prophages detected in whole metaviromes

Original text: As pioneer viruses in the infant gut are thought to be primarily temperate phages induced from the first gut bacterial colonisers²⁰, we next assessed the sharedness of maternal to infant vOTUs while accounting for prophages detected in whole metaviromes.

We thank the reviewer for the comment. This means we expanded our VLP metaviromes with the temperate bacteriophages detected in MGS metaviromes and not concurrent VLP metaviromes (prophages). We have edited the sentence to make it clearer (lines 265-268): *“As pioneer viruses in the infant gut are thought to be primarily temperate phages induced from the first gut bacterial colonisers²⁰, we next assessed the sharedness of maternal to infant vOTUs while accounting for prophages detected only in MGS metaviromes.”*

2. Page 26. Line 556. Seems unusual that 3 out of 4 negative controls failed. Could the authors comment more on this.

It's not uncommon for negative controls to display such behavior, in fact this is totally in line with expectations, especially, when no whole genome amplification was applied to the samples. Negative controls, by design, have minimal or no target microbial reads. As a result, during the PCR amplification within the library preparation process, there might be very little template available for amplification. Hence genomic library preparation was unsuccessful and these samples did not get sequenced. It now reads as: *“Three out of four negative VLP metavirome controls failed library preparation and therefore could not be sequenced <...>”*.

3. Page 30 line 686. Is wilcoxon the proper test here?

As the Shannon Diversity Index is not normally distributed in our dataset, and we took only one time point per individual for this comparison into account (thus accounting for repeated measures is not necessary), the use of Wilcoxon tests is appropriate.

Reviewer #2 (Remarks to the Author):

The Authors have substantially improved the manuscript by increasing the sampling size and frequency, and by improving the data analysis pipeline. While most of the concerns raised in the first round of Review have been addressed, two relevant issues still persist.

- The sequences and metadata are, differently from what stated by the authors in lines 844-845, not publicly available. The access policy associated to the provided EGA

accession says “The Consortium is willing to consider applications from third-party researchers for access to the anonymized sequence data generated by the Project. [...] Access is conditional upon the availability of data and on signed agreement by the researcher to abide by the policies and conditions related to data ownership, disposal, ethical approval, confidentiality and commercialization referred herein”. In fact, nor the fastq files nor the metadata can be downloaded. This is unacceptable. This kind of access restriction, entirely omitted in the manuscript, goes against the Journal Guidelines on data availability and against the FAIR data sharing principles.

Thank you for this very important point. We would like to provide clarity regarding the data sharing procedure. In the previous metagenomics studies of our group, such as Chen et al. (Cell, 2021; doi: 10.1016/j.cell.2021.03.024) and Gacesa et al. (Nature, 2022; doi: 10.1038/s41586-022-04567-7), we always uploaded our data to the European Genome-phenome Archive (EGA). Similarly, for the current study, our dataset — which includes sample information, all phenotypes, family structures, and quality-trimmed sequencing reads — has been securely archived in the EGA under study ID: EGAS00001005969. Our group has a history of fulfilling all data requests. We specifically use EGA for data sharing, as it is an approved repository for functional genomics data according to the Nature Data Repository Guidance. All our prior publications utilized this platform for multi-omic data sharing.

To access our dataset, interested parties must first register with the EGA. After registration, they must read the Data Access form: https://groningenmicrobiome.org/?page_id=2598. This contains a form that requires certain essential details and an acknowledgment from the requester to not use the data for commercial purposes. This form can be found here: <https://forms.gle/zbJLMnojys3VVKPY8>. Requests will be granted for researchers affiliated with an academic, non-profit, or government institution.

In addition to this we have specified this in the text:

Original text: Sample information, basic phenotypes, family structure and quality-trimmed sequencing reads can be found in the EGA repository (study ID: EGAS00001005969).

This has been changed to (lines 848-851): “EGA repository (study ID: EGAS00001005969). Access to the LLNEXT Project data will be granted to qualified researchers will be governed by the provisions laid out in the LLNEXT Data Access Agreement: https://groningenmicrobiome.org/?page_id=2598”

Furthermore, all our contigs have been deposited on figshare and are freely accessible here:

[https://figshare.com/articles/dataset/Virus_scaffolds_reconstructed_in_the_study_b_Mother-](https://figshare.com/articles/dataset/Virus_scaffolds_reconstructed_in_the_study_b_Mother-Infant_Gut_Viruses_and_their_Bacterial_Hosts_Transmission_Patterns_and_Dynamics_during_Pregnancy_and_Early_Life_b_NEXT_Pilot_and_their_metadata_/23926593)

[Infant_Gut_Viruses_and_their_Bacterial_Hosts_Transmission_Patterns_and_Dynamics_during_Pregnancy_and_Early_Life_b_NEXT_Pilot_and_their_metadata_/23926593](https://figshare.com/articles/dataset/Virus_scaffolds_reconstructed_in_the_study_b_Mother-Infant_Gut_Viruses_and_their_Bacterial_Hosts_Transmission_Patterns_and_Dynamics_during_Pregnancy_and_Early_Life_b_NEXT_Pilot_and_their_metadata_/23926593)

- The use of “total metagenomes” vs “bacteriome” and “whole metavirome”, vs “metavirome” vs “virome” is inconsistent throughout the manuscript and the meaning of these terms is not clearly described, nor in the text nor in the figures.

We agree with the reviewer that the use of introduced terms was inconsistent in the original text. For the sake of clarity we have now termed metaviromes as either MGS metaviromes or VLP metavirome. Additionally, we have created a box (Box 1) in the introduction as a glossary of the terms used in this paper. Lastly, we have changed the term microbiome to bacteriome in the text wherever applicable to avoid confusion.

Hereby the text from Box 1:

Bacteriome: community of all bacteria inhabiting an ecosystem

Virome: community of all viruses inhabiting an ecosystem

Total metagenome: the total sum of all genomes of microbes in a sample

VLP metaviromes: the total sum of all genomes of viruses in a VLP-enriched sample

MGS metaviromes: the total sum of all genomes of viruses in a total metagenome

vOTU (virus operational taxonomic unit): species-rank (95% Average Nucleotide Identity, at 85% alignment fraction (relative to the shorter sequence)) virus groups

References

1. Shkoporov, A. N. *et al.* The Human Gut Virome Is Highly Diverse, Stable, and Individual Specific. *Cell Host Microbe* **26**, 527-541.e5 (2019).
2. Shkoporov, A. N. *et al.* Reproducible protocols for metagenomic analysis of human faecal phageomes. *Microbiome* **6**, 68 (2018).
3. Clooney, A. G. *et al.* Whole-Virome Analysis Sheds Light on Viral Dark Matter in Inflammatory Bowel Disease. *Cell Host Microbe* **26**, 764-778.e5 (2019).
4. Garmaeva, S. *et al.* Stability of the human gut virome and effect of gluten-free diet. *Cell Rep.*

- 35**, 109132 (2021).
5. Shkoporov, A. N. *et al.* The Human Gut Virome Is Highly Diverse, Stable, and Individual Specific. *Cell Host Microbe* **26**, 527-541.e5 (2019).
 6. Gulyaeva, A. *et al.* Diversity and Ecology of Caudoviricetes Phages with Genome Terminal Repeats in Fecal Metagenomes from Four Dutch Cohorts. *Viruses* **14**, (2022).
 7. Dahlman, S. *et al.* Temperate gut phages are prevalent, diverse, and predominantly inactive. *bioRxiv* (2023) doi:10.1101/2023.08.17.553642.
 8. Maxwell Anthenelli *et al.* Phage and bacteria diversification through a prophage acquisition ratchet. *bioRxiv* 2020.04.08.028340 (2020) doi:10.1101/2020.04.08.028340.
 9. Howard-Varona, C., Hargreaves, K. R., Abedon, S. T. & Sullivan, M. B. Lysogeny in nature: mechanisms, impact and ecology of temperate phages. *ISME J.* **11**, 1511–1520 (2017).
 10. Roux, S., Enault, F., Hurwitz, B. L. & Sullivan, M. B. VirSorter: mining viral signal from microbial genomic data. *PeerJ* **3**, e985 (2015).
 11. Paul, J. H. Prophages in marine bacteria: dangerous molecular time bombs or the key to survival in the seas? *ISME J.* **2**, 579–589 (2008).
 12. Song, W. *et al.* Prophage Hunter: an integrative hunting tool for active prophages. *Nucleic Acids Res.* **47**, W74–W80 (2019).
 13. Akhtar, M. M., Micolucci, L., Islam, M. S., Olivieri, F. & Procopio, A. D. Bioinformatic tools for microRNA dissection. *Nucleic Acids Res.* **44**, 24–44 (2016).
 14. Arndt, D., Marcu, A., Liang, Y. & Wishart, D. S. PHAST, PHASTER and PHASTEST: Tools for finding prophage in bacterial genomes. *Brief. Bioinform.* **20**, 1560–1567 (2019).
 15. Hockenberry, A. J. & Wilke, C. O. BACPHLIP: predicting bacteriophage lifestyle from conserved protein domains. *PeerJ* **9**, e11396 (2021).
 16. Shang, J., Tang, X. & Sun, Y. PhaTYP: predicting the lifestyle for bacteriophages using BERT. *Brief. Bioinform.* **24**, bbac487 (2023).
 17. Roux, S. *et al.* iPHoP: An integrated machine learning framework to maximize host

prediction for metagenome-derived viruses of archaea and bacteria. *PLOS Biol.* **21**, e3002083 (2023).

18. Anneliek M. ter Horst, Jane D. Fudyma, Jacqueline L. Sones, & Joanne B. Emerson. Dispersal, habitat filtering, and eco-evolutionary dynamics as drivers of local and global wetland viral biogeography. *bioRxiv* 2023.04.28.538735 (2023)

doi:10.1101/2023.04.28.538735.

19. Peng, Y. *et al.* Viruses in deep-sea cold seep sediments harbor diverse survival mechanisms and remain genetically conserved within species. *ISME J.* **17**, 1774–1784 (2023).

20. Sáenz, J. S., Rios-Galicia, B., Rehkugler, B. & Seifert, J. Dynamic Development of Viral and Bacterial Diversity during Grass Silage Preservation. *Viruses* **15**, (2023).

REVIEWERS' COMMENTS

Reviewer #1 (Remarks to the Author):

Major Comments:

1. The fact that this is an analysis of only dsDNA virome should be more prominently noted such as in the abstract etc.

2. "This difference in sharedness may be attributed to the fact that our study encompassed longitudinal samples from both mother and infant for a longer time frame after birth, whereas Maqsood et al. looked at the sharing of vOTUs between mother and infant cross-sectionally around birth which would mostly represent vertical transmission viruses during birth".

One of the other major differences between this paper and others, is the use of VLP plus metagenomic sequencing without viral prep. What is the overlap between the VLP virome and the MGS virome? This is important to share as many groups are mining metagenomic data for virome analysis.

Reviewer #2 (Remarks to the Author):

The Authors have addressed all concerns raised in the previous round of Review.

My only recommendation is to fix the "run_sample.csv" metadata file, as part of the file is indeed comma separated, while part is separated by semi-columns, and add the corresponding column headers rather than having all the relevant metadata collapsed in the "sample_attributes" field.

REVIEWERS' COMMENTS

Reviewer #1 (Remarks to the Author):

Major Comments:

1. The fact that this is an analysis of only dsDNA virome should be more prominently noted such as in the abstract etc.

Original text (lines 20-22): “We longitudinally assessed the composition of gut viruses and their bacterial hosts in 322 total metagenomes and 205 Virus Like Particle (VLP) metaviromes from 30 mothers during and after pregnancy and from their 32 infants during their first year of life.”

We agree with the Reviewer and have edited the abstract accordingly (lines 4-8): “To study the development of the infant gut virome over time and the factors that shape it, we longitudinally assess the composition of gut viruses and their bacterial hosts in 30 women during and after pregnancy and in their 32 infants during their first year of life. Using shotgun metagenomic sequencing applied to dsDNA extracted from Virus-Like Particles (VLPs) and bacteria, we generate 205 VLP metaviromes and 322 total metagenomes.”

We then mention dsDNA gut virome in 205 VLP metaviromes as early as in the first sentence of Results (line 63-66): “We profiled the gut microbiome (primarily referred to as the bacteriome) in 322 total metagenome samples and the double-strand DNA (dsDNA) gut virome in 205 VLP metavirome samples from 30 mothers and their 32 term-born infants (including 2 twin pairs) collected longitudinally from pregnancy to 12 months after birth (Fig. 1a; Supplementary Fig. 1a, b).”

2. “This difference in sharedness may be attributed to the fact that our study encompassed longitudinal samples from both mother and infant for a longer time frame after birth, whereas Maqsood et al. looked at the sharing of vOTUs between mother and infant cross-sectionally around birth which would mostly represent vertical transmission viruses during birth”.

One of the other major differences between this paper and others, is the use of VLP plus metagenomic sequencing without viral prep. What is the overlap between the VLP virome and the MGS virome? This is important to share as many groups are mining metagenomic data for virome analysis.

We agree with the Reviewer completely and now report the overlap between the VLP metaviromes and MGS metaviromes in Methods section “Generation of MGS metaviromes”, lines 657-664:

“The average alignment rate from total metagenomes to the curated virus database was $50.6 \pm 18.2\%$. As for VLP metaviromes, a count table was generated and transformed for MGS metaviromes. For samples where both VLP and MGS metaviromes were available (204 out of 322), the median Bray-Curtis distance between VLP and MGS metaviromes of the same faecal sample comprised 0.86 ± 0.20 . This distance to the own concurrent sample was lower than to samples of unrelated individuals that averaged at 0.99 ± 0.05 (p-value < 0.001, one-sided Wilcoxon rank sum test run through 1,000 permutations).”

However, it is important to bear in mind that the reported overlap is estimated using the concurrent MGS and VLP metaviromes. To truly estimate the efficacy of mining viruses solely from total metagenomes, it is necessary to run additional analyses involving VLP-independent reconstruction of virus genomes from total metagenomes and compare those to virus genomes reconstructed from concurrent VLP metaviromes.

Reviewer #2 (Remarks to the Author):

The Authors have addressed all concerns raised in the previous round of Review. My only recommendation is to fix the "run_sample.csv" metadata file, as part of the file is indeed comma separated, while part is separated by semi-columns, and add the corresponding column headers rather than having all the relevant metadata collapsed in the "sample_attributes" field.

We agree with the Reviewer on that matter, however, the EGA web platform transforms metadata files submitted as a plain text file into SQL-like files for efficient parsing and searching within the whole archive. We have, therefore, created a script that reformats run_sample.csv back into the plain text with a tab separator and parses all relevant metadata in the respective columns. The script is accessible from the GitHub repository of the paper: [https://github.com/GRONINGEN-MICROBIOME-CENTRE/LLNEXT_pilot/blob/main/Data Access EGA.md](https://github.com/GRONINGEN-MICROBIOME-CENTRE/LLNEXT_pilot/blob/main/Data%20Access%20EGA.md)